# Photosynthesis Performance at Different Growth Stages, Growth, and Yield of Rice in Saline Fields

**DOI:** 10.3390/plants12091903

**Published:** 2023-05-07

**Authors:** Supranee Santanoo, Watanachai Lontom, Anoma Dongsansuk, Kochaphan Vongcharoen, Piyada Theerakulpisut

**Affiliations:** 1Department of Biology, Faculty of Science, Khon Kaen University, Khon Kaen 40002, Thailand; supranee4705@hotmail.com (S.S.); watalo@kku.ac.th (W.L.); 2Department of Agronomy, Faculty of Agriculture, Khon Kaen University, Khon Kaen 40002, Thailand; danoma@kku.ac.th; 3Faculty of Science and Health Technology, Kalasin University, Kalasin 46000, Thailand; ko-cha_9@hotmail.com

**Keywords:** biomass, chlorophyll fluorescence, KDML105, net photosynthesis rate, *Oryza sativa* L., Pokkali, saline soils

## Abstract

Photosynthetic performance and biomass at different growth stages of the salt-sensitive KDML105 rice cultivar, three improved lines (RD73, CSSL8-94, and TSKC1-144), and the salt-tolerant standard genotype (Pokkali) were investigated under non-saline, semi-saline, and the heavy-saline field conditions in the northeast of Thailand. In the non-saline field, net photosynthesis rates (P_n_) of all genotypes remained high from the early vegetative stage to the milky stage and then dramatically reduced at maturity. In contrast, in both saline fields, P_n_ was the highest at the early vegetative stage and continuously declining until maturity. Leaf chlorophyll content remained high from the early vegetative to milky stage then reduced at maturity for all three field conditions. During the reproductive phase, P_n_ of KDML105 and the improved lines were reduced by 4–17% in the heavy-saline field, while that of Pokkali was increased (11–19% increase over that of the non-saline). Pokkali also showed a prominent increase in water use efficiency (WUE) under salinity. Nevertheless, rice leaves under saline conditions maintained the PSII integrity, as indicated by the pre-dawn values of maximum quantum yield of PSII photochemistry (F_v_/F_m_) of higher than 0.8. Pokkali under the semi-saline and the heavy-saline conditions exhibited 51% and 27% increases in final biomass, and 64% and 42% increases in filled grain weight plant^−1^, respectively. In the semi-saline condition, RD73, TSKC1-144, CSSL8-94, and KDML105 showed moderate salt tolerance by displaying 24%, 18.6%, 15%, and 11.3% increases in final biomass, and 24%, 4%, 13%, and 6% increases in filled grain weight plant^−1^, respectively. In contrast, in the heavy-saline field, final biomass of RD73, KDML105, CSSL8-94, and TSKC1-144 showed 48%, 45%, 38%, and 36% reductions from that in the non-saline field, while the filled grain weight plant^−1^ were reduced by 45%, 58%, 35%, and 32%, respectively. This indicated that the improved lines carrying drought- and/or salt-tolerance genes achieved an increased salt tolerance level than the parental elite cultivar, KDML105.

## 1. Introduction

Rice (*Oryza sativa* L.) has the third largest planting area in the world with a 10-year average, from 2011 to 2020, of 16.25 million ha, while yield production per area was 4545 kg ha^−1^ [1]. Rice is one of the main sources of daily food for more than 50% of the world’s population, and is cultivated in more than 100 countries with 90% of the total global production from Asia [2]. For Thailand, rice is the most important economic crop with the planting area of 10.82 million ha (yield production per area 3032 kg ha^−1^) and more than half of rice production for domestic consumption and export comes from the northeast (NE) region [3]. Paradoxically, while the NE accounts for the largest rice-growing area (e.g., 61% or 5.76 million ha in 2018 wet season), yield per area in this region has always been the lowest. For instance, rice yields in the wet season of 2018 were recorded at 3912, 3606, 3000, and 2225 kg ha^−1^ in the Central, Northern, Southern and NE, respectively [4]. The yield gap in the country was quite large compared to the world’s potential yield per area, especially in the NE region. The most economically important rice cultivars in the northeast, Khao Dawk Mali 105 (KDML105) and RD6, are sensitive to abiotic stresses, and are usually seriously damaged from abiotic stress leading to reduced income of poor farmers.

During the last few decades, climate change has become a serious threat to agriculture and caused significant production loss of a wide variety of crops [5,6,7]. In dry and semi-arid areas, soil salinity is the major environmental constraint limiting plant productivity [8]. In the NE region of Thailand, only 10% of rice fields are under irrigation, while almost 90% are rain-fed lowland paddy fields. In addition to poor soils and drought, rice yields are also severely reduced or totally lost when cultivated in saline soils. Salt-affected areas in the NE are formed through geochemical and anthropogenic process, and cover the area of 1.84 million ha, accounting for 85% of country salinity areas (21.62 million ha). Salinity levels of soils in this region are classified into slightly (approximately area 0.016 million ha), moderately (0.036 million ha), and severely (1.17 million ha) affected based on electrical conductivity of saturated soil extract (EC_e_) [9]. Salinity is a soil condition characterized by a high soluble salt concentration. Soils having an EC_e_ of 4 dS m^−1^, equivalent to approximately 40 mM NaCl generating an osmotic pressure of approximately 0.2 MPa, are classified as being saline. Soils having an EC_e_ greater than 4 dS m^−1^ were reported to cause significant yield reduction in most crops [10]. Among cereal crops, rice has been grouped as the salt susceptible cereal with a threshold of 3 dS m^−1^ for most cultivated varieties, and 12% yield reduction was recorded for an increase in 1 unit of EC_e_ [11]. The response of rice under salt stress is complex and depends on the type of salt stress, duration of salt exposure, and growth stage [12]. According to previous reports, soils with conductivities higher than 4 and 8 dS m^−1^ are considered, respectively, moderate and highly saline for rice crops [13,14]. Soil at low EC_e_ of 3.5 dS m^−1^ resulted in 10% yield loss, and 50% loss was recorded at EC_e_ 7.2 dS m^−1^ [15]. However, Asch and Wopereis [16] reported that the flood water with the EC level of more than 2 dS m^−1^ caused the yield loss of up to 1 t ha^−1^ per unit EC (dS m^−1^) in rice. Moreover, Hussain et al. [12] reported that the low EC_e_ at 1 dS m^−1^ (1.5 g NaCl kg^−1^ soil) reduced plant height and root length as well as dry matter content of stem, root, and total dry weight in early seedling stage and maximum tillering stage of rice. Lutts et al. [17] reported that rice is most sensitive to salt stress at the seeding and early vegetative stage, and later at the reproductive stage [18,19]. The excess salt in soils adversely affects plant growth, development, and productivity when osmotic stress reduces water uptake by roots [10]. Accumulation of Na^+^ to toxic levels in plant tissues disturbs metabolic processes, particularly photosynthesis, and all major morpho-physiological and yield-related traits, including tiller number, panicle length, spikelet number per panicle [20], grain filling [21], and plant biomass [22], leading to significantly decreased yield.

Photosynthesis is the primary determinant of biomass and yield, and numerous CO_2_ enrichment studies provided compelling evidence that yield of many crops, including rice, can be increased by increasing net photosynthesis rate (P_n_) [23,24]. Photosynthesis is sensitive to abiotic stress, including salinity. The immediate effects of salinity on photosynthetic ability of rice involved an induction of stomatal closure, due to salt-induced osmotic stress, which restricted CO_2_ diffusion and directly reduced net photosynthesis rate [25]. Accumulation of toxic Na^+^ and Cl^−^ ions in rice tissues after an extended period of salt stress resulted in a reduction in P_n_ from multiple factors including a reduction in chlorophyll content, loss of membrane integrity and electrolyte leakages, damaged chloroplast ultrastructure, reduced effective quantum yield of PSII photochemistry (ΦPSII) and electron transport rate (ETR), inhibition of Calvin cycle enzymes, and reduced P_n_ [26,27,28]. Salinity also disturbs normal reactive oxygen (ROS) homeostasis, leading to oxidative stress due to the over-production of ROS, particularly in the chloroplast [29]. Previous studies conducted mostly under greenhouse conditions reported that salt-tolerant rice genotypes were better able than the salt-sensitive ones in maintaining chlorophyll content [27], ΦPSII and ETR [30], P_n_ [28], and efficient antioxidant systems [26]. However, photosynthesis performance of rice under saline field conditions has rarely been reported. Many crop scientists believed that enhancing photosynthesis at the level of single leaf would contribute to yield increment [31]. The level of salt tolerance of rice varied with growth stages; therefore, investigation of photosynthesis performance at different growth stages is important for prediction of growth and of rice under the saline fields.

According to previous experiments in our research group, greenhouse-grown salt-sensitive KDML105 rice challenged with moderate salt stress exhibited lower photosynthesis performance compared with the salt-tolerant standard rice genotypes, such as Pokkali or DH103 [28,32]. Some improved KDML105 rice lines, such as CSSL8-94 and RGD4, were derived from marker-assisted backcross breeding (MABC), in which drought- or salt-tolerance quantitative trait loci (QTL) were introgressed into the KDML105 genome, also showed higher photosynthesis performance than KDML105 [28]. The differential physiological parameters, such as photosynthesis, could serve as important tools for the selection criteria of rice breeders for future breeding schemes. Therefore, it would be important to confirm the differential photosynthesis performance under field conditions to make the best use of this trait for breeding, as well as modifications of cultural practice for improvement of yield production of salt sensitive KDML105 rice. The objectives of this study were to investigate growth and photosynthesis performance at different growth stages, and yield of saline-field-grown rice genotypes, including KDML105, three improved lines (RD73, CSSL8-94, TSKC1-144), compared with the standard salt-tolerant variety, Pokkali. This information will help determine genotypic potential and/or environmental factors limiting photosynthetic performance, growth, and yield of rice.

## 2. Results

### 2.1. Environmental Conditions in the Area

Total rainfall, number of rainy days, monthly mean relative humidity (RH), and air temperature (T) of each month in 2021 are presented in Figure 1. During the hot season (March and April), prior to the rice growing season, the mean of monthly total rainfall was 114.1 mm, rainy days was 7.5 days, minimum T was 24.4 °C, maximum T was 34.6 °C, and RH was 68.8% (Figure 1A–C and Appendix A). The rainy season (May to October) experienced a mean of monthly total rainfall of 147.2 mm, rainy days of 15.5 days, and RH of 78.5%. The shaded area in Figure 1 represented the climatic conditions during the planting period (27 July to 11 November 2021) which showed the highest rainfall (271.5 mm), number of rainy days (23), and RH (86.8%) in September. During the pre-planting period in the hot season (March, 2021), soil electrical conductivity (EC) of heavy-saline, semi-saline, and non-saline plot were measured as 45.27, 13.45, and 0.86 dS m^−1^, respectively (Appendix A). During the planting season (July to November), the soil EC dramatically reduced due to the flood water in the experimental plots and the heavy rainfall. As noted in Figure 1D, during July–September, EC of the flood water in the heavy-saline (1.48–2.97 dS m^−1^) and the semi-saline plot (1.31–2.26 dS m^−1^) were higher than that of the soil (1.12–1.33 and 1.05–1.21 dS m^−1^, respectively). During October–November, after the flood water was drained out from the experimental plots, soil EC of the heavy-saline, the semi-saline, and the non-saline plots dramatically increased showing the mean values of 4.29 (with the maximum of 6.5), 3.66 (maximum, 4.83), and 0.67 (maximum, 0.88) dS m^−1^, respectively (Figure 1D; Appendix A).

### 2.2. Photosynthetic Performance

Photosynthesis parameters, including leaf chlorophyll content, P_n_, gs, T_r_, and WUE of five rice genotypes at six different growth stages, are displayed in Figure 2 and Appendix A. Three-way ANOVA (Table 1) revealed that all five photosynthesis parameters among genotypes and growth stages are highly significantly different (*p* < 0.001), but all except P_n_ are significantly different (*p* < 0.001) among saline fields. Moreover, the interactions among the three factors caused highly significantly differences (*p* < 0.001) in all five leaf gas exchange parameters. The P_n_ of rice plants growing in the non-saline field were moderate at the early vegetative stage (mean across genotypes was 23.29 µmol CO_2_ m^−2^ s^−1^), slightly increased with age, and then abruptly decreased to 14.08 µmol CO_2_ m^−2^ s^−1^ at maturity (Figure 2D). A similar trend was observed in leaf chlorophyll content (Figure 2A). In contrast, P_n_ of rice growing in the semi-saline and heavy-saline fields were highest at the early vegetative stage (mean across genotypes were 28 µmol CO_2_ m^−2^ s^−1^ for both saline fields), then continuously decreased with age (Figure 2E,F) despite the stability of leaf chlorophyll content (SPAD values) from the early vegetative stage to the milky stage (Figure 2B,C). The mean P_n_ across genotypes of the mature plants were 14.08, 15.95, and 11.12 µmol CO_2_ m^−2^ s^−1^ for the non-saline, semi-saline, and heavy-saline plants, respectively (Figure 2D–F; Appendix A). It was interesting to note the contrasting response to high salinity between the salt-tolerant genotype, Pokkali and the remaining genotypes. In Pokkali, P_n_ during reproductive stages were higher in the heavy-saline than the non-saline field while the opposite occurred in other genotypes. For Pokkali, P_n_ at the early booting stage increased from 22.26 in the non-saline field to 26.01 µmol CO_2_ m^−2^ s^−1^ in the heavy-saline field (16% increase), at flowering stage from 19.40 to 23.09 µmol CO_2_ m^−2^ s^−1^ (19% increase), and at milky stage from 18.29 to 20.36 µmol CO_2_ m^−2^ s^−1^ (11% increase) (Appendix A). Contrastingly, in the remaining genotypes, such as RD73, P_n_ decreased from 27.98 in non-saline to 24.82 µmol CO_2_ m^−2^ s^−1^ in the heavy-saline (11% reduction) during early booting, from 26.58 to 23.37 µmol CO_2_ m^−2^ s^−1^ (12% reduction) at flowering, and from 25.63 to 19.15 µmol CO_2_ m^−2^ s^−1^ (25% reduction) at milky stage (Appendix A).

In the non-saline field, rice plants at the early vegetative stage had much lower gs (mean across genotypes of 0.12 mol H_2_O m^−2^ s^−1^) than those in the saline fields which were 0.48 and 0.47 mol H_2_O m^−2^ s^−1^, for semi-saline and heavy-saline fields, respectively (Figure 2G–I). The large difference in gs could be due to different time of measurement, i.e., the values in the non-saline field were recorded around 11:30 am while those in the saline fields were performed before 10:00 am, reflecting the effects of temperature, humidity, and light intensity on stomatal conductance. However, the opposite was observed at the stem elongation stage, i.e., mean gs across genotypes of plants in the non-saline field increased to the maximum of 0.53 mol H_2_O m^−2^ s^−1^ (Figure 2G), while the mean gs for plants in the saline fields dramatically reduced to approximately 0.15 mol H_2_O m^−2^ s^−1^ (Figure 2H,I). The large difference was related to the effects of both salinity and the environment during measurement. At the early booting and flowering stages, gs of plants in the saline fields increased to the similar levels as those in the non-saline field, then declined toward maturity. During the reproductive stages (early booting to maturity), mean gs across genotypes in the heavy-saline were lower than those in the non-saline and the semi-saline fields which indicated the effects of soil salinity on stomatal closure. Moreover, genotypic differences in gs in the saline fields were apparent, particularly the gs values of Pokkali were significantly (*p* < 0.05) lower than those of the other genotypes during the early booting and flowering stages (Figure 2H,I; Appendix A). The patterns of change in transpiration rates (T_r_) with plant developmental stages were more or less similar to changes in gs (Figure 2J–L). The WUE of the non-saline plants was highest at the early vegetative stage, abruptly dropped at the early booting, then continuously increased until the milky stage, and finally dropped to a minimum at maturity (Figure 2M). In contrast, WUE of rice grown in saline fields was low at the early vegetative stage, increased to maximum at the stem elongation stage, then dropped and stabilized until the milky stage (Figure 2N,O). The most clearly observed differences among genotypes were found in the case of Pokkali at early booting which had significantly (*p* < 0.01) higher WUE than other genotypes in the saline fields (Figure 2N,O; Appendix A).

Chl fluorescence parameters, including effective quantum yield of PSII photochemistry (ΦPSII), maximum quantum yield of PSII photochemistry in the light (F_v_′/F_m_′) and electron transport rate (ETR) of the five rice genotypes at six different growth stages are displayed in Figure 3 and Appendix A. Three-way ANOVA (Table 1) revealed that all three Chl fluorescence parameters are significantly different (*p* < 0.001) among saline fields and growth stages, while ΦPSII and ETR showed no differences (*p* > 0.05) among genotypes. However, only the interaction between saline fields and growth stages had significant effects on all three Chl fluorescence parameters. At the early vegetative stage, the mean ΦPSII across genotypes of rice leaves growing in the non-saline and semi-saline fields were highest at 0.31 and 0.30, respectively (Figure 3A,B). At later growth stages, mean ΦPSII across genotypes of plants in the non-saline and semi-saline plots significantly and continuously declined until maturity. For the heavy-saline field, the maximum mean ΦPSII across genotypes was found at the stem elongation (0.33) before being reduced with age until maturity. At maturity, the mean ΦPSII across genotypes of rice in the heavy-saline (0.14) was significantly (*p* < 0.05) lower than that of the non-saline plants (0.16). The mean F_v_′/F_m_′ across genotypes at the early vegetative stage was highest in the semi-saline (0.58), followed by that of the heavy-saline (0.54) and the non-saline (0.50) plants (Figure 3D–F; Appendix A). In most cases, no significant differences among genotypes were observed, except for Pokkali growing in the heavy-saline field which showed significantly (*p* < 0.01) higher ΦPSII (0.34) than other genotypes (0.26–0.29) at the early booting stage. The patterns of changes in the mean F_v_′/F_m_′ across genotypes were similar for the semi- and heavy-saline plants, i.e., the efficiency increased with age to a maximum at the flowering stage (0.61 and 0.59 for the semi- and heavy-saline, respectively), then declined at later stages (Figure 3E,F). In contrast, the mean F_v_′/F_m_′ across genotypes of the non-saline plants increased to a maximum (0.55) at the early booting stage before slowly declining until maturity (Figure 3D). At maturity, the mean F_v_′/F_m_′ across genotypes was highest in the semi-saline (0.47) followed by the heavy-saline (0.43) and lowest in the non-saline (0.37) plants. The patterns of change in ETR (Figure 3G–I) were similar to those of ΦPSII (Figure 3A–C). The mean ETR across genotypes were highest in the heavy-saline plants at the stem elongation stage (174.87 µmol e^−^ m^−2^ s^−1^). Meanwhile, the maximum ETR of the non-saline (167.28 µmol e^−^ m^−2^ s^−1^) and semi-saline (159.76 µmol e^−^ m^−2^ s^−1^) plants were observed at the early vegetative stage. It was noted that, at stem elongation, early booting, and flowering stages, the mean ETR across genotypes in the heavy-saline field were higher than the semi-saline and non-saline fields (Appendix A).

### 2.3. Diurnal Chlorophyll Fluorescence

Diurnal Chl fluorescence under ambient light intensity was measured from 05:30 to 17:30 at the stem elongation, the flowering, and the mature stages corresponding to the middle of rainy season, end of rainy season, and the beginning of cool season, respectively, and the results are displayed in Figure 4, Figure 5 and Figure 6. In general, diurnal patterns of leaf temperature and ETR followed the unimodal pattern of light intensity, while the ΦPSII showed the opposite pattern. At the stem elongation stage, leaf temperatures at dawn (05:30) were approximately 25 °C in all three fields (Figure 4D–F). Temperatures of rice leaves in the two saline fields reached the maximum earlier, at 11:30, while those in the non-saline field peaked at 13:00. Leaf temperatures were continuously declining during the afternoon reaching a temperature slightly lower than that in the early morning. The ΦPSII measured at 5:30 which represented the maximum quantum yield of PSII photochemistry in the dark-adapted state (F_v_/F_m_) were higher than 0.8 in all three fields, indicating healthy leaf status (Figure 4G–I). During the morning hours, the ΦPSII values were decreasing with increasing light intensity to reach the minimum at 11:30, the peak of light intensity, for rice plants in the non-saline and semi-saline fields (the means across genotypes were 0.14 and 0.23, respectively). In contrast, the ΦPSII of rice plants in the heavy-saline condition reached the minimum as early as 8:30. After midday, the ΦPSII values were increasing with decreasing light intensity to reach the values slightly higher than 0.8 at 17:30 which were close to the F_v_/F_m_ values at dawn. Different diurnal patterns among rice fields were also observed for ETR. The non-saline plants showed the maximum ETR as early as 8:30 (the mean ETR across genotypes was 144.01 µmol e^−^ m^−2^ s^−1^) after which the ETR continuously decreased until 17:30 (except for Pokkali and TSKC1-144). These genotypes exhibited a bimodal pattern of changes in ETR showing a smaller peak at 13:30 (Figure 4J). The ETR of rice plants in the two saline fields, on the other hand, reached the maximum at 11:30 coinciding with the light intensity peak. The mean ETR across genotypes at 11:30 were 161.92 and 170.29 µmol e^−^ m^−2^ s^−1^ for the semi- and heavy-saline plants, respectively (Figure 4K,L).

The measurements at the flowering stage were conducted on flag leaves at the end of rainy season 21 days after water drainage from the rice fields and soil EC were highest. The temperatures of flag leaves at dawn (05:30) were approximately 21 °C in all three fields which were approximately 4 °C lower than that at the stem elongation stage (Figure 5D–F). As shown in Figure 5D,E, leaf temperatures of the rice plants in the non-saline and semi-saline plots reached the maximum at 11:30 coinciding with the peak of light intensity. However, leaf temperature of plants in the heavy-saline field increased earlier and reached the maximum at 10:00 (Figure 5F). The rates of leaf temperature reduction during the afternoon were slower in the semi- and heavy-saline than the non-saline plants. Unlike the situation at the stem elongation stage, leaf temperatures at 17:30 were approximately 5–6 °C higher than that at dawn. The ΦPSII measured at 5:30 were higher than 0.8 in all three fields (Figure 4G–I). The ΦPSII values of rice plants in the two saline fields decreased to the minimum earlier at 10:00, while that of the non-saline plants fell to a minimum at 11:30. The rate of increment in ΦPSII during the afternoon was much slower in the heavy-saline than in the semi- and non-saline fields. Unlike the situation at the stem elongation stage, the ΦPSII values at 17:30 were lower than 0.8. The ETR of rice in the non-saline and semi-saline fields reached the peaks at 10:00 (Figure 5J,K) with the mean ETR across genotypes of 173.07 and 205.27 µmol e^−^ m^−2^ s^−1^, respectively. However, rice plants in the heavy-saline field attained the maximum ETR (242.14 µmol e^−^ m^−2^ s^−1^) later at 11:30. Genotypic differences were noted in the semi-saline field in that KDML105 tended to have higher ETR than others at 10:00, and in the heavy-saline field Pokkali, KDML105, and CSSL8-94 had higher ETR than others at 11:30.

The patterns of diurnal changes in flag leaf temperature at the mature stage (Figure 6D–F) were similar to those at the flowering stage except that the maximum leaf temperatures were attained later in the afternoon and the means across genotypes were lower because the measurements were conducted in the early cool season. The ΦPSII measured at 5:30 were higher than 0.8 in all three fields (Figure 4G–I). The minimum ΦPSII in all three plots occurred at 10:00 coinciding with the peak of light intensity. It was noted that the minimum ΦPSII values in the heavy-saline (0.09 to 0.28) were lower than those in the non-saline (0.14 to 0.29) and the semi-saline (0.23 to 0.38) plants. The ETR in the non-saline fields varied a lot depending on genotypes. In the semi-saline field, Pokkali showed the higher maximum ETR (277.94 µmol e^−^ m^−^^2^ s^−^^1^) at an earlier time (10:00) than other genotypes. In the heavy-saline field, TSKC1-144 showed the earliest (8:30–10:00) and highest ETR (196.50 µmol e^−^ m^−^^2^ s^−^^1^) compared to other genotypes.

### 2.4. Growth and Biomass

Changes in plant height, biomass of leaves, stems, roots, and total plant dry weight at different rice growth stages are displayed in Figure 7; Appendix A. Three-way ANOVA (Table 2) showed that all five growth and biomass parameters are highly significantly different (*p* < 0.001) among saline fields and growth stages. However, root dry weights are not significantly different (*p* > 0.05) among genotypes, and genotypes x saline fields interaction. Salinity caused significant (*p* < 0.01) reductions in plant height. At maturity, the mean plant height across genotypes in the non-, semi-, and heavy-saline fields were 166.2, 146.95, and 136.15 cm, respectively, with Pokkali and TSKC1-144 as the tallest genotypes. In the heavy-saline field, the mean of total dry weight at maturity across five genotypes was 87.94 g plant^−1^ which was 33% reduced from that in the non-saline field (130.86 g plant^−1^). In contrast, in the semi-saline field, the mean of total dry weight at maturity across five genotypes (159.78 g plant^−1^) increased 22% over that in the non-saline field. However, patterns of changes in biomass at maturity differed among genotypes. In the non-saline field, total dry weights at maturity of four rice genotypes (RD73, CSSL8-94, TSKC1-144, and KDML105) varied from 134 to 150 g plant^−1^ (Figure 7M), while dry biomass in the heavy-saline field reduced to 71–85 g plant^−1^ (Figure 7O). In contrast, total dry biomass of Pokkali increased from 92.78 g plant^−1^ in the non-saline to 139.98 g plant^−1^ (51% increase) in the semi-saline, and 117.08 g plant^−1^ (26% increase) in the heavy-saline field. In the semi-saline field, plant growth tended to be stimulated, resulting in an increase in total dry weight at maturity for all genotypes (Figure 7N). In the semi-saline condition, RD73, TSKC1-144, CSSL8-94, and KDML105 showed moderate salt tolerance by displaying 24%, 18%, 15%, and 11% increase in biomass, respectively, compared to dry weights in the non-saline field. In contrast, total biomass of all four genotypes in the heavy-saline field significantly decreased (−48%, −45%, −38%, and −36% for RD73, KDML105, CSSL8-94, and TSKC1-114, respectively). Based on total dry biomass at maturity, all three improved lines performed better than KDML105 under the semi-saline condition. However, in the heavy-saline field, only CSSL8-94 and TSKC1-144 were more salt tolerant than KDML105.

The pattern of changes in root biomass differed between non-saline and saline fields (Figure 7J–L). In both saline fields, root biomass continuously increased until maturity for all genotypes (except for TSKC1-144 in the semi-saline field). However, in the non-saline field, Pokkali and KDML105 attained the highest root biomass (26.44 and 31.46 g plant^−1^, respectively) at stem elongation, then declined at later stages. The other three genotypes had the highest root biomass at flowering, and then declined toward maturity. Stem biomass for non-saline plants of all genotypes were maximum at flowering then declined toward maturity (Figure 7G). In the semi-saline field, stem biomass continuously increased until maturity, with TSKC1-144 having significantly (*p* < 0.05) higher stem biomass (48.20 g plant^−1^) than other genotypes (24.16–30.38 g plant^−1^) (Figure 7H). The pattern of changes in stem biomass in the heavy-saline field was similar to that in the non-saline plot, except for Pokkali which had maximum stem dry weight at the stem elongation stage (21.17 g plant^−1^), then remained stable until maturity (Figure 7I). The changes in leaf dry weight during development varied widely among genotypes and field conditions (Figure 7D–F). Generally, leaf biomass was maximized at the stem elongation or flowering stage, then declined at maturity. An interesting pattern was noted for KDML105 and CSSL8-94 growing in the semi-saline field in which their leaf biomass continuously increased toward maturity (Figure 7E). Among genotypes, TSKC1-144 showed the most prominent growth in the semi-saline field, while Pokkali tended to be more robust than others in the heavy-saline condition. In the non-saline field, on the other hand, KDML105 showed the highest growth potential.

### 2.5. Sodium (Na^+^) and Potassium (K^+^) Content and Yield

Two-way ANOVA (Table 3) indicated that salinity strongly affected Na^+^ and K^+^ contents and K^+^/Na^+^ ratio (*p* < 0.001) in rice tissues, while differences among genotypes are significant only for Na^+^ content (*p* < 0.001) and K^+^/Na^+^ ratio (*p* < 0.05). The Na^+^, K^+^, and K^+^/Na^+^ ratio of the above ground tissues of mature plants of five rice genotypes growing under non-saline, semi-saline, and heavy-saline field conditions are presented in Table 4. As shown in Table 4, for all rice genotypes, Na^+^ contents under saline conditions significantly (*p* < 0.01) increased compared with the those under the non-saline field. The mean Na^+^ contents across genotypes was the highest in the heavy-saline (0.028%), followed by the semi-saline (0.023%) and the non-saline plants (0.007%). The significant difference (*p* < 0.01) in Na^+^ content among rice genotypes was observed only in the semi-saline plot. The Na^+^ contents of KDML105 (0.030%), and CSSL8-94 (0.030%) were significantly higher than Pokkali (0.014%) and TSKC1-144 (0.016%), while RD73 (0.024%) showed the intermediate value. The K^+^ contents among rice genotypes in the non-saline condition were significantly different (*p* < 0.01), while no significant differences were observed in the saline plants. The K^+^ contents in the non-saline plants were highest in TSKC1-144 (0.952%) and Pokkali (0.852%) which were significantly different from RD73 (0.732%), CSSL8-94 (0.655%), and KDML105 (0.650%). For the K^+^/Na^+^ ratios, the significant differences among rice genotypes were observed in the non- and semi-saline plants, while the differences were not significant for the heavy-saline plants. The tolerant genotypes, Pokkali and TSKC1-144, had significantly higher K^+^/Na^+^ ratio than RD73, CSSL8-94, and KDML105 in both non- and semi-saline plot (Table 4). The large increase in the mean Na^+^ across genotypes in the heavy-saline plants, i.e., 4.0 times compared to that in the non-saline field, led to a 5.88-fold reduction in K^+^/Na^+^ ratio.

Seven rice yield production parameters, including panicle length, total grain number per panicle, number of filled and unfilled grains per panicle, filled grain weight per panicle, filled grain weight per plant, and weight of 100 filled grains are shown in Table 5, and the morphology of rice panicles are displayed in Figure 8. Two-way ANOVA (Table 3) revealed that salinity had significant (*p* < 0.05) effects on four yield characters but did not affect (*p* > 0.05) total grain number panicle^−1^, number of filled grains panicle^−1^, and number of unfilled grains panicle^−1^. Significant differences (*p* < 0.05) among genotypes were found only for panicle length, number of unfilled grains panicle^−1^, and weight of 100 filled grains. As can be seen in Figure 8, panicle length and the number and length of the primary branches were similar for the non-saline (Figure 8A–E) and semi-saline (Figure 8F–J) plants. However, strong salinity imposed negative effects on these characters of the heavy-saline plants (Figure 8K–O). As shown in Table 5, the mean of panicle length across genotypes in the heavy-saline plants (26.7 cm) was significantly (*p* < 0.05) lower than those in the semi-saline (28.6 cm) and non-saline (29.1 cm) ones. Only RD73 and TSKC1-144 had significantly (*p* < 0.01) decreased panicle length in the heavy-saline condition (Table 5). The significant differences (*p* < 0.05) among rice genotypes were observed only in the non-saline plants, i.e., the panicle length of TSKC1-144 (33.7 cm) and RD73 (30.7 cm) were significantly higher than CSSL8-94 (25.4 cm), while Pokkali (28.9 cm) and KDML105 (26.9 cm) had intermediate length. Across the five genotypes, rice plants growing in the semi-saline field produced higher number of total (186 grains panicle^−1^) and filled grains (154 grains panicle^−1^) than the non-saline plants, and the lowest numbers were recorded in the heavy-saline field, although the differences were not significant. In the two saline fields, no significant differences among genotypes were recorded in both total and filled grain number panicle^−1^. The only significant differences among genotypes were found in the non-saline plants with TSKC1-144 producing the highest grain number (175 grains panicle^−1^) which was significantly higher than KDML105 (127 grains panicle^−1^) and Pokkali (106 grains panicle^−1^). Across the five genotypes, no significant differences in the number of unfilled grains panicle^−1^ were found among the three fields. The mean of filled grain weight panicle^−1^ across genotypes were similar for the non-saline (4.19 g panicle^−1^) and semi-saline plants (4.16 g panicle^−1^) and significantly (*p* < 0.05) higher than that of the heavy-saline plants (3.21 g panicle^−1^). Among genotypes, the difference in filled grain weight panicle^−1^ was observed only in the non-saline plants ranging from 5.08 (TSKC1-144) to 3.42 (Pokkali) g panicle^−1^. It is interesting to note that the mean of filled grain weight plant^−1^ across genotypes was the highest in semi-saline plants (25.86 g plant^−1^) followed by the non-saline (21.68 g plant^−1^), and the heavy-saline (14.89 g plant^−1^). In the semi-saline condition, KDML105, TSKC1-144, CSSL8-94, RD73, and Pokkali produced 4%, 6%, 13%, 24%, and 64% increase in filled grain weight plant^−1^ compared to that in the non-saline field. In the heavy-saline condition, KDML105, RD73, CSSL8-94, and TSKC1-144 showed 58%, 45%, 35%, and 32% reduction, respectively, in filled grain weight plant^−1^ compared to those in the non-saline field. In contrast, filled grain weight plant^−1^ of Pokkali growing in the heavy-saline field showed a 42% increase compared to the non-saline field. Although in the non-saline condition the three improved lines produced lower filled grain weight plant^−1^ (21.10–25.27 g) than KDML105 (25.36 g), they all had higher grain weight in the heavy-saline field, indicating the higher salt tolerance ability. The mean of the 100-grain weight across genotypes was the greatest in non-saline (3.03 g) followed by the semi- (2.71 g) and the heavy-saline (2.59 g) plants, indicating adverse effects of salinity on the grain-filling process. Similar trends were observed in all genotypes. In comparison with the non-saline plants, the percentage reduction in the 100-grain weight was lowest in Pokkali (6%) and highest in RD73 (21%) and KDML105 (19%) while those of CSSL8-94 and TSKC1-144 were reduced by 11% and 15%, respectively (Table 5).

### 2.6. Principal Component Analysis (PCA) and Hierarchical Cluster Analysis (HCA) for Biomass, Yield, and Physiological Parameters

For the clear visualization of relationships between rice genotypes growing in different saline fields and growth, yield, and physiological attributes, two-way hierarchical cluster analysis (HCA) and principal component analysis (PCA) were carried out and their results displayed in Figure 9. The heatmap clearly divided rice into two major clusters based on similarity in physiological responses and agronomic characters at maturity (Figure 9A). All five rice genotypes growing in the non-saline plot are grouped together in cluster A within which KDML105, CSSL8-94, RD73, and TSKC1-144 are grouped in A1. Pokkali growing under non-saline and both saline conditions showed similar responses, indicating its high salt tolerance ability and are, therefore, clustered together in one group (A2). The second major cluster (B) consists of KDML105, CSSL8-94, and RD73 in semi-saline (B3); KDML105, CSSL8-94, and RD73 in heavy-saline field (B2); and TSKC1-144 in both saline conditions (B1). This indicated that among the three improved lines TSKC1-144 performed differently, under salinity, compared to CSSL8-94 and RD73 which showed similar response patterns to KDML105. Discriminative parameters for better performance of TSKC1-144 included K^+^ content, plant height, stem biomass, and LAI. Physiological and agronomical characters which tended to be high under non-saline conditions and greatly reduced in the semi- and heavy-saline fields are shown in cluster II including K^+^ content, K^+^/Na^+^ ratio, plant height, and weight of 100 grains, while high WUE is the prominent discriminative characters for Pokkali. Parameters in cluster Ia and Ib tended to be high under non-saline and semi-saline and greatly reduced in the heavy-saline field. Characters which were clearly enhanced to a great extent by the slightly saline soil condition included leaf chlorophyll content, Fv’/Fm’, and root biomass (Ic). To a lesser extent, net photosynthesis rate (Ia), tiller number (Ib), and total biomass (Ib) were also enhanced in the semi-saline field. The prominent characters which differentiated TSKC1-144 from CSSL8-94, RD73, and KDML105 under semi-saline condition were higher K^+^, plant height, and stem biomass.

The results of the PCA obtained from data of agro-physiological parameters at maturity of five rice genotypes growing under the three saline conditions are illustrated in Figure 9B. The first three components explained 74.42% of the total variation between parameters. The 3-D plot of the three components clearly separated Pokkali growing in the three fields into one group. KDML105, RD73, CSSL8-94, and TSKC1-144 in the non-saline field are grouped together. Under semi-saline condition, the responses of all five genotypes differed widely. However, in the heavy-saline field, KDML105, CSSL8-94, and RD73 showed similar responses and are grouped together. On the other hand, TSKC1-144 under the heavy-saline showed distinctive responses compared to other genotypes.

## 3. Discussion

The ability of rice to tolerate salt-induced damage to photosynthesis varied widely depending on genotypes, salinity level, and growth stage [33]. At a moderate level of salt stress (EC of 6 dS m^−1^), P_n_ of flag leaves reduced from the controls by 47% in the salt-sensitive rice IR29 but only 17% in the salt-tolerant IR632 [26]. At high salinity (14.8 dS m^−1^) after 10 days of salt stress, P_n_ at vegetative stage reduced by 51% in the salt-tolerant RGD4, and 68% in salt-sensitive KDML105 [28]. Numerous reports on photosynthesis in rice under salt stress in pot or hydroponic cultures [26,27,28,32] are available but observations in saline fields [34] are scarce. In this study, changes in P_n_ with development clearly differed between rice growing in the non-saline and the saline fields. During the vegetative phase (early vegetative and stem elongation), P_n_ in both saline fields were not reduced, or even stimulated, compared to that in the non-saline plot (Figure 2D–F) confirming previous reports that rice during vegetative growth is relatively tolerant to salinity [35]. During reproductive phase from early booting to the milky stage, P_n_ of rice genotypes in the non-saline remained high (Figure 2D) while the rates in the heavy-saline field had a tendency to reduce with age. Moreover, P_n_ at each stage in the heavy-saline field was slightly lower than that in the non-saline (except Pokkali). This result corroborated the observations that rice is most sensitive to salt stress at the reproductive stage leading to yield reduction because of poor pollination, fertilization, and grain filling due to reduced photosynthesis of source leaves [18,36].

Contrasting responses of photosynthesis performance were clearly evident between the salt tolerant Pokkali and the less-tolerant remaining genotypes. Compared to the non-saline condition, P_n_ of Pokkali in the heavy-saline field exhibited 16%, 19%, and 11% increase at the early booting, flowering, and milky stage, respectively (Appendix A). In contrast, P_n_ of the remaining genotypes in the heavy-saline field during these stages were, in most cases, reduced from those in the non-saline field (Appendix A). Therefore, the soil EC between 2.64–6.50 dS m^−1^ (Figure 1D) during reproductive stages were high enough to cause significant accumulation of toxic Na^+^ and severe reduction in K^+^/Na^+^ (Table 4) resulting in metabolic disturbance, including a reduction in P_n_ of KDML105 and the three improved lines at the milky to mature stage (Figure 2; Appendix A). The reduction in P_n_ was affected by high Na^+^ and was known to be attributed to (i) stomatal limitation following stomatal closure (reduced gs as defense mechanism to reduce water loss) leading to limited CO_2_ diffusion into the leaf, and (ii) non-stomatal limitations which included a reduction in mesophyll conductance, inhibition of activity of CO_2_ fixation enzymes, damaging photosynthetic apparatus, reduction in PSII photochemical efficiency, obstruction of electron transport, disruption of membrane transport regulation, and oxidative damage due to accumulation of reactive oxygen species [28,37,38,39,40]. For the less tolerant genotypes (RD73, KDML105, CSSL8-94, and TSKC1-144), the reduction in P_n_ was mainly caused by stomatal limitations considering the parallel trend in the reduction in gs and P_n_ (Appendix A). In contrast, the enhancement of photosynthesis of Pokkali under the heavy-saline condition (at early booting and flowering), despite the huge reduction in gs (Figure 2I), could be attributed to the good maintenance of leaf water status following significantly lower T_r_ (Figure 2L) and prominently high WUE (Figure 2O). A previous study in pot cultures found that the net photosynthesis rate of Pokkali was not reduced by salinity, while that of CSR-13 (another salt-tolerant genotype) was significantly reduced when subjected to salt stress at 12 dS m^−1^ [41]. This work confirmed that the photosynthesis ability of Pokkali was tolerant of salt stress. Munns and James [42] suggested that low stomatal conductance and high WUE were among the most effective screening characteristics for salinity tolerant genotypes.

Although leaf chlorophyll content of rice plants in the saline fields increased with age from the early vegetative to milky stage (Figure 2B,C), the P_n_ continuously declined during this phase (Figure 2E,F). Moreover, rice leaves in both saline fields at all growth stages had significantly higher SPAD values than those in the non-saline condition, while P_n_ in most cases were lower (except Pokkali) (Appendix A). Therefore, the salinity levels in these fields (i.e., lower than 6 dS m^−1^, Appendix A) had no negative effects on chlorophyll biosynthesis but obstructed photosynthesis performance. Increased chlorophyll content under moderate salt stress (EC of 6 dS m^−1^) was also observed in pot-grown rice seedlings subjected to stress for 10 days [43]. However, previous studies in pot or hydroponic culture reported that salinity caused a significant reduction on SPAD values even in salt-tolerant genotypes, such as Pokkali [28]. It should be noted that, in most greenhouse experiments, higher salt concentrations were used, and plants could be under less favorable micro-environments than in the field.

In saline fields, the patterns of changes in ΦPSII and ETR (Figure 3B,C,H,I) were similar to those of P_n_ (Figure 2E,F). Flexas et al. [44] reported that under mild water stress the decrease in P_n_ and ETR was proportional, but under severe stress ETR remained high while P_n_ dramatically reduced. Under abiotic stress conditions combined with high light intensity, ETR was up-regulated while CO_2_ fixation was down-regulated. Under stress, electron transport through the non-cyclic pathway, which directly produces NADPH and ATP for Calvin cycle slowed down, and extra electrons were channeled to several alternative pathways (such as Mehler reaction and cyclic electron flow through PGR5/PGRL1 or NDH-1 proteins) to alleviate photoinhibition of PSI [45]. It is worth noting in this study that ETR at stem elongation and flowering stage in the heavy-saline were much higher than those in the non-saline field while P_n_ in the heavy-saline were not different from that in the non-saline field (Appendix A) indicating that alternative electron transport pathways were accelerated under heavy-saline conditions to help prevent photoinhibition.

Although salinity induced a reduction in P_n_, particularly in genotypes with lower salt tolerance level, the maximum PSII photochemical efficiency measured in the dark (F_v_/F_m_) at 5:30 remained high (>0.8) throughout development, in both non-saline and saline fields, from stem elongation, flowering, and even maturity (Figure 4, Figure 5 and Figure 6). This high pre-dawn F_v_/F_m_ values indicated that the leaves did not suffer from chronic damages to thylakoid photosynthetic apparatus which will cause an inactivation of PSII [46,47]. The diurnal patterns of changes in PSII photochemical efficiency (ΦPSII) in this study were similar to that observed in rice in a greenhouse study [48], and in field-grown rice [49]. At the stem elongation stage, with low EC level in the saline fields, the rate of afternoon recovery of ΦPSII were similar in all three fields (Figure 4), and at 17:30 PSII efficiency fully recovered to the same level as the pre-dawn values. However, at flowering stage coinciding with the highest soil EC levels (Figure 1D), the rate of afternoon recovery of PSII efficiency in the heavy-saline (Figure 5I) tended to be slower than that in the non-saline field (Figure 5G), and at 17:30 the ΦPSII values were lower than those at pre-dawn. It was noted that the slower rate of afternoon recovery of PSII efficiency in the heavy-saline field coincided with the slower rate of reduction in leaf temperature (Figure 5F). Slower recovery of ΦPSII was also observed in *Populus euphratica* under more stressful drought conditions [50]. The PSII efficiency was also affected by plant age. In rice, the lowest midday ΦPSII, as well as rate of afternoon recovery were reducing with advanced developmental stages [48].

Salinity hampers growth of rice, and percentage growth reduction can be used to classify salt tolerance levels of rice genotypes [43]. In this study, the plant height, total biomass, as well as biomass in different plant parts (leaf, stem, and root) of all rice genotypes (except Pokkali) in the heavy-saline were significantly lower than those in the non-saline field (Appendix A). Growth reduction in the heavy-saline field was attributed to an inhibition of cell division and enlargement due to osmotic and ion toxicity effects, and inadequate photo-assimilate supply [10], resulting in the lowest tiller number (Appendix A), and, consequently, the lowest total dry weight in all plant parts (Appendix A) and the lowest yield and yield components (Table 5). Negative effects of soil EC on tiller number, biomass, and yield were reported in both greenhouse [18] and saline paddy fields [51,52]. Although Pokkali growing in the non-saline field had the lowest total biomass at maturity among genotypes, its total biomass in the heavy-saline field became the highest. This striking stimulation of growth under salinity, i.e., 26% increased final biomass of Pokkali under the heavy-compared to that in the non-saline field (Appendix A) could be attributed, primarily, to the increase in P_n_ during booting (16%), flowering (19%), and milky stages (11%) as mentioned above. Moreover, the increase in leaf biomass in the heavy-, compared with the non-saline field (Appendix A), which reflected increased photosynthetic leaf area at stem elongation (+14%) and flowering (+7%) contributed to an increase in canopy photosynthesis which cumulatively resulted in increased final total biomass at maturity. In addition, higher root biomass in the heavy-saline plants (Figure 7L) also helped Pokkali maintain vigorous shoot growth, probably through salt dilution and salt exclusion during uptake, resulting in limited accumulation of toxic Na^+^ in the shoot [43]. Furthermore, it was well-documented that salt-tolerant rice genotypes such as Pokkali had higher expression and more efficient functioning of the HKT1;5 transporter protein, encoded by the *SKC1* gene [53], than that in salt-sensitive genotypes. This transporter protein which mostly located in cell membrane of root parenchyma cells played a prominent role in salt exclusion by up-taking Na^+^ out from root xylem vessels thereby minimizing transport of toxic Na^+^ into shoots [54,55,56]. In the semi-saline plot, total biomass of KDML105 and the three improved lines at flowering were much reduced compared to those in the non-saline field (Appendix A), coinciding with the reduction in P_n_ (Appendix A) because of increased soil EC (Figure 1D). In contrast, biomass at maturity of these genotypes in the semi-saline were significantly higher than those in the non-saline condition (Appendix A). This can be explained by growth stimulation during 2–3 weeks before maturity which occurred due to heavy rainfall for 2 days in a row at around the milky stage leading to a huge reduction in soil EC (Figure 1D). For all rice genotypes, the low soil EC during the milky stage to maturity (1.28–4.05 dS m^−1^; Figure 1D) also resulted in greater filled grain weight plant^−1^ in the semi-saline compared to that in the non- and heavy-saline fields (Table 5). Therefore, growth and grain yield of rice under saline field conditions varied widely depending not only on the soil properties but also the climatic conditions, especially the precipitation [51,57].

The results from this experiment confirmed the beneficial effects, on growth under salinity, of the introgression of DT-QTL8 and *SKC1* gene into KDML105 genome. In comparison to KDML105, in the heavy-saline field, the introgression lines CSSL8-94 (carrying DT-QTL8) and TSKC1-144 (carrying both DT-QTL8 and *SKC1*) showed higher total biomass at maturity, and also lower percentage reduction compared to those in the non-saline field (Appendix A). In the semi-saline field, all three improved lines showed higher percentage increase in total biomass at maturity than KDML105 (Appendix A). Beneficial effects of the introgressed genes were also expressed in yield parameters. In comparison to KDML105, in the heavy-saline field, all three improved lines showed higher filled grain weight plant^−1^ and lower percentage reduction compared to that in the non-saline field (Table 5). Moreover, with regard to the 100-filled grain weight under the heavy-saline condition, TSKC1-144 grains were significantly heavier than KDML105 (Table 5). The higher salt tolerance of CSSL8-94 and RD73 over that of KDML105 has previously been reported in Pamuta et al. [58] for growth attributes, and in Suriyaaroonroj et al. [59] for growth and yield. In the heavy-saline field, TSKC1-144, in comparison with CSSL8-94 and RD73, exhibited higher plant height, higher biomass of leaf and root at maturity, (Appendix A) as well as higher total grain number panicle^−1^ and higher 100-filled grain weight (Table 5). This indicated the additive effects of pyramiding the drought- and salt-tolerant genes in TSKC1-144.

The PCA based on physio-agronomic parameters clearly differentiated Pokkali from the other genotypes (Figure 9). Pokkali was minimally affected by both salinity levels and, therefore, Pokkali under non-saline, semi-saline, and heavy-saline conditions were clustered together. The improved cultivar RD73 which was introgressed with the major QTL, Saltol (containing *SKC1* gene regulating Na^+^-K^+^ homoeostasis), under semi-saline field conditions, exhibited significantly lower Na^+^ content, slightly higher K^+^/Na^+^ (Table 4), slightly higher total stem biomass (Appendix A), and slightly higher filled grain plant^−1^ weight (Table 5). On the other hand, the line CSSL8-94 (introgressed with DT-QTL8) did not show any improved characters in relation to Na^+^-K^+^ homoeostasis, but had slightly higher biomass (Appendix A), slightly higher total grain number panicle^−1^, and slightly filled grain weight plant^−1^ (Table 5), compared to KDML105. Hence, considering multiple physio-agronomic parameters, PCA resulted in clustering CSSL8-94 and RD73 in the same group as KDML105, under both saline environments. Interestingly, TSKC1-144 which contained both *SKC1* gene and DT-QTL8, had low Na^+^ and high K^+^/Na^+^ (significantly different from KDML105 and comparable to Pokkali; Table 4). In addition, TSKC1-144 had a significantly higher weight of 100 filled grains, slightly higher biomass, slightly higher total grain number panicle^−1^, and filled grain weight plant^−1^. Considering overall physio-agronomic response traits, PCA for both saline conditions resulted in a separation of TSKC1-144 from KDML105, CSSL8-94, and RD73 (Figure 9). Therefore, TSKC1-144 is a valuable genetic resource to be subjected to further genetic improvement to obtain a potential new cultivar for enhanced salt tolerance in the field conditions.

The saline experimental plots used in this study had very high soil EC in the pre-planting hot season (27.3 to 57.7 dS m^−1^ for the heavy-saline; 6.56 to 13.11 dS m^−1^ for the semi-saline; Appendix A) with thick salt crust on the soil surface, and the farmers avoided growing rice in these areas because rice plants were not able to survive beyond the vegetative phase [51,57]. This study showed that, with proper water and cultural management, elite rice cultivar, such as KDML105, could survive until maturity in the area of high salinity with 45% reduction in final biomass at maturity (Appendix A) and 58% reduction in filled grain weight plant^−1^ (Table 5), compared to those in the non-saline condition. Flooding the soils to 30 cm above soil surface using non-saline irrigation water for several days before transplanting and maintaining water level until the end of vegetative phase effectively diluted salinity to the levels between 1.21 and 2.97 dS m^−1^ (Appendix A). It was suggested that standing water in lowland rice throughout the season was the best option to remove salt from root zone through leaching and dilution [60,61]. In addition, in this study, the use of 40-day-old rice seedlings for transplanting helped increase the chance of survival, compared to the 30-day-old seedlings used in conventional rice growing by local farmers. Moreover, in this study, transplanting later in the mid-rainy season, around two weeks later than the conventional practice of farmers in this area, led to a better survival of seedlings due to salt dilution by heavy rain. Therefore, proper management of water, soil, and cultural practice together with the use of suitable rice genotypes are recommended for farmers to ensure survival and better growth of rice in salt-affected land.

## 4. Materials and Methods

### 4.1. Site Description and Microclimate

The field experiments were conducted at Daeng Yai, Mueang District, Khon Kean province, in Northeastern Thailand (16.49° N, 102.73° E, altitude 195 m above sea level). The characteristic of the climate is a tropical savanna according to Köppen’s classification [62]. The weather conditions of this area varied between seasons, with rainy season lasting about six months (May to October), cold and dry season for four months (November-February), and hot season for two months (March to April). During 2009–2019, the annual number of rainy days was 111 days, with the mean total rainfall being 1314 mm, while maximum and minimum mean air temperatures were 34.40 and 20.89 °C, respectively [63]. In 2021, the monthly meteorological conditions including total rainfall and number of rainy days, relative humidity (RH), maximum and minimum air temperature near the field site (~7 km) were obtained from the Upper Northeastern Meteorological Center, Mueang District, Khon Kean province during January to December 2021.

The soil used in this experiment is Kula Ronghai series (Ki: fine-loamy; mixed, active, isohyperthermic Typic Natraqualfs) [64]. In this study, rice was grown in three different plots, including heavy-saline (salinity > 8 dS m^−1^), semi-saline (3–5 dS m^−1^) and non-saline (<2 dS m^−1^) (Appendix A). The non-saline plot was located approximately 300 m from the non-saline one but on a higher ground of about 3 m. For each plot, soil samples prior to planting were taken from four points at depths of 0 to 15 cm. The following physical and chemical properties were determined based on [65], i.e., percentages of sand, silt and clay, organic matter, organic carbon, total nitrogen, available phosphorus (P), and exchangeable potassium (K). The soil textures were loam with low total nitrogen and organic matter contents. Other physical and chemical soil properties of each plot at pre-planting were analyzed and presented in Table 6.

### 4.2. Plant Materials and Cultural Practice

Five rice genotypes, including Pokkali, KDML105, RD73, CSSL8-94, and TSKC1-144, were used for this experiment. Pokkali is a standard salt tolerance cultivar, while KDML105 is the world-renown Thai aromatic rice but sensitive to salinity [66]. The remaining genotypes were cultivar or lines improved from KDML105 via marker-assisted backcross selection (MABC). RD73 is a new cultivar released by the Thailand Rice Department in 2017 recommended for growing in saline soils; it has KDML105 genetic background introgressed with *SKC1* (salt tolerance gene on chromosome 1) [66]. CSSL8-94 is a chromosome segment substitution line (CSSL) developed from KDML105 introgressed with drought tolerance QTL located on chromosome 8 (DT-QTL8) [67]. TSKC1-144 is a backcross breeding line with KDML105 genetic background resulting from a cross between CSSL8-94 and RGD4 (KDML105 introgressed with *SKC1*), therefore *SKC1* and DT-QTL8 were pyramided in this line [68].

Rice seeds were obtained from Rice Science Center and Rice Gene Discovery Unit, Kasetsart University (KDML105, CSSL8-94, and Pokkali), Nakorn Ratchasima Rice Research Center (RD73), and Salt-tolerant Rice Research Group, Khon Kaen University (TSKC1-144). Seed germination started on 18 June 2021, and seedlings were grown in cement blocks (1 m × 5 m) filled with paddy soils in a net house at Khon Kaen Rice Research Center. After 40 days, they were uprooted and transplanted into each of the 6.5 m × 11.75 m ploughed field (non-saline, semi-saline, and heavy-saline). The experimental design in each field was a randomized complete block design (RCBD) with four replications. Each replication contained 5 subplots (1.25 m × 1.75 m) representing 5 rice genotypes. Each subplot contained 35 plants (5 plants × 7 rows) at 25 × 25 cm spacing.

Land preparation was carried out by following the normal procedures for experimental fields of rice. The fields were flooded to 30 cm above soil surface from the time of transplanting (28 July 2021) to the early booting stage when the flooded water was drained out on 3 Oct 2021. EC of water of each plot was measured by using conductivity meter (Mettler Toledo FiveEasy™ pH/mV bench meter model F20, Mettler-Toledo, Inc., Switzerland). The ECs of the soils were determined by measuring the EC of saturated soil extracts (1:5 soil:water) [69] The NPK complex fertilizers were applied at the tillering stage (30 days after transplanting, DAT) and panicle initiation stage (60 DAT) at the rate based on soil analysis and nutrient recommendation for rice [70]. Weeds were manually removed throughout the experimental fields.

### 4.3. Photosynthetic Performance

#### 4.3.1. Leaf Chlorophyll Content, Leaf Gas Exchange and Chlorophyll Fluorescence

Leaf chlorophyll content, leaf gas exchange, and Chl fluorescence were measured at six growth stages, i.e., early vegetative, stem elongation, early booting, flowering, milky, and mature. For each growth stage and genotype, data were collected from one plant in each replication (n = 4). On each plant, each parameter was measured at the middle of the second fully expanded leaf for plants at the vegetative stages (early vegetative and stem elongation) and the flag leaf for those at the reproductive stages (early booting, flowering, milky, and mature). Leaf chlorophyll content was determined using chlorophyll meter (Minolta SPAD-502 Plus, Konica Minolta Inc., Osaka, Japan). Leaf gas exchange and Chl fluorescence data were collected during 9:00 to 11:30 am using an infrared gas analyzer (IRGA) model Li-cor 6400xt with an LED light source (6400-02B Red/Blue Light Source, Li-Cor Inc., Lincoln, NE, USA). The conditions during measurements were controlled as follows, photosynthetically active radiation (PAR) at 1200 μmol photon m^−2^ s^−1^, CO_2_ concentration at 400 μmol mol^−1^, and temperature at 30 ± 2 °C. Leaf gas exchange parameters measured included net photosynthesis rate (P_n_), stomatal conductance (gs), transpiration rate (T_r_) and water-use efficiency (WUE). The WUE was calculated as the ratio of P_n_/T_r_. Chl fluorescence parameters reported included effective quantum yield of PSII photochemistry (ΦPSII), maximum quantum yield efficiency of PSII in the light (F_v_′/F_m_′), and electron transport rate (ETR). The ETR was calculated by the equation: ETR = ΦPSII × 0.84 × 0.5 × PAR [47].

#### 4.3.2. Diurnal Chlorophyll Fluorescence

Patterns of changes in diurnal Chl fluorescence were measured at three growth stages, including stem elongation, flowering, and maturity. For each genotype, Chl fluorescence parameters were measured on two plants for each genotype in each field using Mini PAM-II Photosynthesis Yield Analyzer (Heinz Walz GmbH, Effeltrich, Germany). Minimal fluorescence yield of the dark-adapted state (F_0_) was measured in complete darkness before sunrise (05:30). Maximal fluorescence of the dark-adapted state (F_m_) was obtained following a saturating pulse of 5000 μmol photon m^−2^ s^−1^ lasting 0.8 s. The maximum photochemical quantum yield of PSII (F_v_/F_m_) was calculated using the formular F_v_/F_m_ = (F_m_ − F_0_)/F_m_. Steady state fluorescence in the light-adapted state (F′) and the maximal fluorescence of the light-adapted state (F_m_′) were measured during the day between 07:00–17:30 (every 1.5 h, 8 time points). ΦPSII value was calculated from the equation, ΦPSII = (F_m_′ − F′)/F_m_′ [46,71].

### 4.4. Growth, Biomass and Yield

Rice seeds were germinated on 18 June 2021, transplanted in the flooded fields on 28 July 2021 at the seedling age of 40 d (the number of leaves was 3–4, and the height was 40–45 cm). Before transplanting, leaf lamina were excised and discarded from the tip to approximately two-thirds of the lamina length to minimize transpirational loss. It took approximately two and four weeks after transplanting for the non-saline and saline fields, respectively, for the seedlings to completely recover and started to produce new leaves. Growth and photosynthesis data were collected at six growth stages. The first data collection was carried out when the plants produced 2–3 fully expanded leaves in the main stem and 2–3 young tillers (~17 days after transplanting (DAT) for the non-saline field, and ~35 DAT for the saline fields), and this growth stage will be referred to as ‘early vegetative growth’. Data collection for the second growth stage was carried out at ‘stem elongation’ (~50 DAT for the non-saline field, and ~57 DAT for the saline fields) when the plants attained maximum tiller numbers before panicle initiation occurred. The third growth stage for data collection was carried out at ‘early booting’ (~67 DAT for the non-saline field, and ~74 DAT for the saline fields). The fourth growth stage for data collection was at ‘flowering’ (~80 and ~85 DAT), when approximately 50% of the main stem and tillers exerted their panicles. The fifth stage of data collection was performed at the ‘milky stage’ (~85 DAT for the non-saline field, and ~90 DAT for the saline fields). Finally, data were collected from plants at ‘mature stage’ (~105 DAT for the non-saline field, and ~107 DAT for the saline fields).

Plant height was measured at all six growth stages from five randomly selected plants from each replication (*n* = 20). Above and below ground biomass was recorded at four growth stages, i.e., early vegetative growth, stem elongation, flowering, and mature, from one plant in each replication (*n* = 4). Rice roots were sampled with the monolith, size of soil volume approximately 1000 cm^−3^ (10 cm wide, 10 cm long and 10 cm depth, with the plant stem located In the center. The leaves, stems, and roots were detached from each plant, dried at 80 °C for 48 h (BF 720, INDER GmbH (Headquarters), Tuttlingen, Germany) or until weight was constant and then weighed to obtain dry weight. At maturity, sodium (Na^+^), and potassium (K^+^) content of dried aboveground samples (0.1 g) was determined using an atomic absorption spectroscopy (Corning, Model GBC932AAA, UK) after digesting in 10 mL of nitric acid at 300 °C followed by 5 mL perchloric acid at 200 °C, and 20 mL of 6 M hydrochloric acid.

Yield and yield components were determined at maturity including panicle length, total grain number per panicle, number of filled and unfilled grains per panicle, filled grain weight per panicle and per plant, and weight of 100 filled grains. The data of each parameter were obtained from the main panicle of three plants in each replication (n = 12).

### 4.5. Data Analysis

Data normality was tested using Shapiro–Wilk [72] in the Statistix version 10 software (Analytical Software, Tallahassee, FL, USA). According to data distribution, the results were subjected to three- and two-way ANOVA for assessing the significance of quantitative changes in various parameters due to different rice genotypes and saline fields at different growth stages. The Tukey’s honestly significant difference test (HSD) [73] were used for multiple comparisons of means at an alpha level of 0.05. All statistical analyses were taken by using Sigmaplot Version 11.0 software (San Jose, CA, USA,) and followed the procedure described by Gomez and Gomez [74]. Principal component analysis (PCA) was used to analyze and determine the relationships between physiological parameters, growth, and yield. Hierarchical cluster analysis (HCA) with a heatmap was used to group rice genotypes growing under different saline fields based on photosynthetic performance, growth, and yield data. Pearson’s correlation, PCA and HCA were conducted using OriginPro 2022 software (OriginLab Corporation, Northampton, MA, USA).

## 5. Conclusions

The responses of five rice genotypes in this study showed a wide variation of physio-agronomic responses under non-, semi-, and heavy-saline fields. The standard salt tolerant Pokkali, in both saline fields, displayed better photosynthesis performance, greater biomass, and higher yield traits (number of filled grain panicle-1, filled grain weight panicle^−1^, and filled grain weight plant^−1^) than those in the non-saline field. In contrast, all physiological and agronomic parameters of the salt-sensitive elite cultivar (KDML105) and three improved genotypes, were negatively affected under the heavy-saline field, while responses in the semi-saline field greatly varied. Among the three improved genotypes, TSKC1-144 (introgressed with a salt-tolerant gene, SKC1, and a drought-tolerant QTL) displayed greater salt tolerance than RD73 (containing only SKC1) and CSSL8-94 (carrying only the drought-tolerant QTL). These improved lines proved to be good potential sources of breeding materials for further backcross breeding to obtain salt tolerant rice with excellent cooking quality, such as KDML105.

## Figures and Tables

**Figure 1 plants-12-01903-f001:**
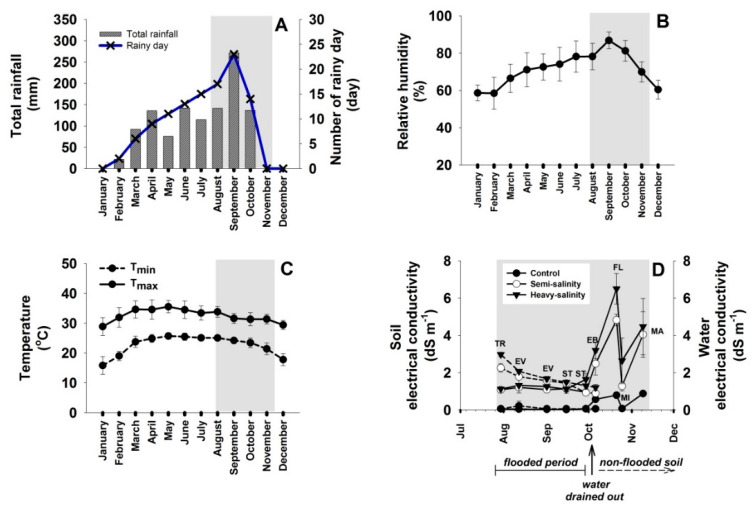
The meteorological parameters including total rainfall and number of rainy days (**A**), monthly mean relative humidity (**B**), and minimum and maximum temperature (**C**) during January to December 2021 were recorded by Upper Northeastern Meteorological Center, Mueang Khon Kaen District, Khon Kaen. Soil (**—**) and water (**---**) electrical conductivity (EC) of non-saline (control), semi-saline, and heavy-saline plot (**D**) were measured at the time of transplanting (TR), early vegetative growth (EV), stem elongation (ST), early booting stage (EV), flowering (FL), milky (MI), and mature stage (MA).

**Figure 2 plants-12-01903-f002:**
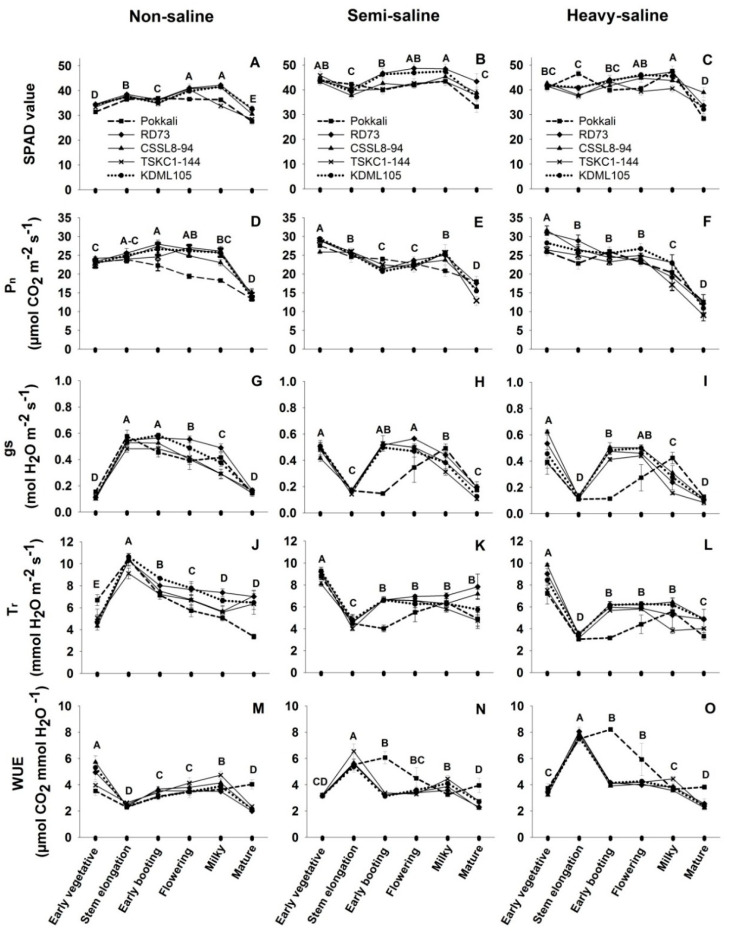
Photosynthetic parameters including SPAD values (**A**–**C**), net photosynthesis rate (P_n_, (**D**–**F**)), stomatal conductance (gs, (**G**–**I**)), transpiration rate (T_r_, (**J**–**L**)), and water–use efficiency (WUE, (**M**–**O**)) of five rice genotypes (■, Pokkali; ◆, RD73; ▲, CSSL8–94; **X**, TSKC1–144 and ●, KDML105) at six growth stages, including early vegetative, stem elongation, early booting, flowering, milky, and mature stage. Rice plants were grown under non-saline, semi-saline, and heavy-saline field conditions. Each capital letter at each growth stage represents the mean of five genotypes. Different capital letters indicated significant (*p* < 0.05) differences among different growth stages.

**Figure 3 plants-12-01903-f003:**
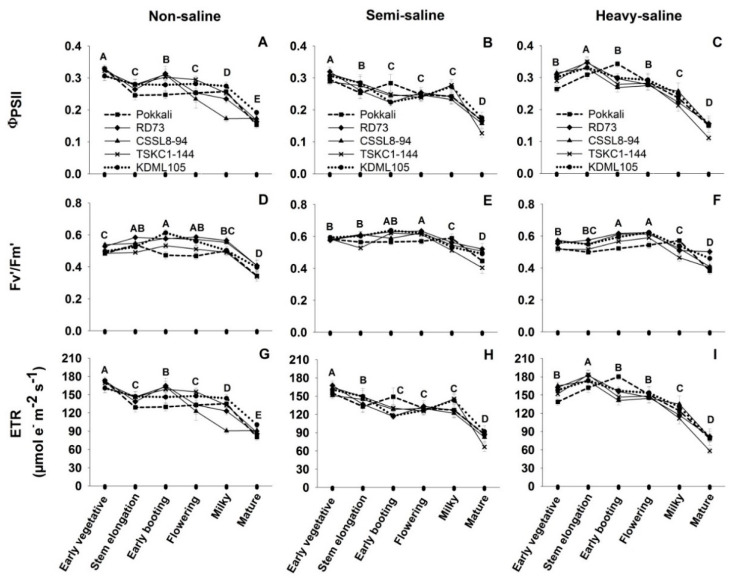
Chlorophyll fluorescence parameters including effective quantum yield of PSII photochemistry (ΦPSII, (**A**–**C**)), maximum quantum yield efficiency of PSII (F_v_′/F_m_′, (**D**–**F**)) and electron transport rate (ETR, (**G**–**I**)) of five rice genotypes (■, Pokkali; ◆, RD73; ▲, CSSL8–94; **X**, TSKC1–144; and ●, KDML105) at six growth stages including early vegetative, stem elongation, early booting, flowering, milky, and mature stage. Rice plants were grown in non-saline, semi-saline, and heavy-saline fields. Each capital letter at each growth stage represents the mean of five genotypes. Different capital letters indicated significant (*p* < 0.05) differences among different growth stages.

**Figure 4 plants-12-01903-f004:**
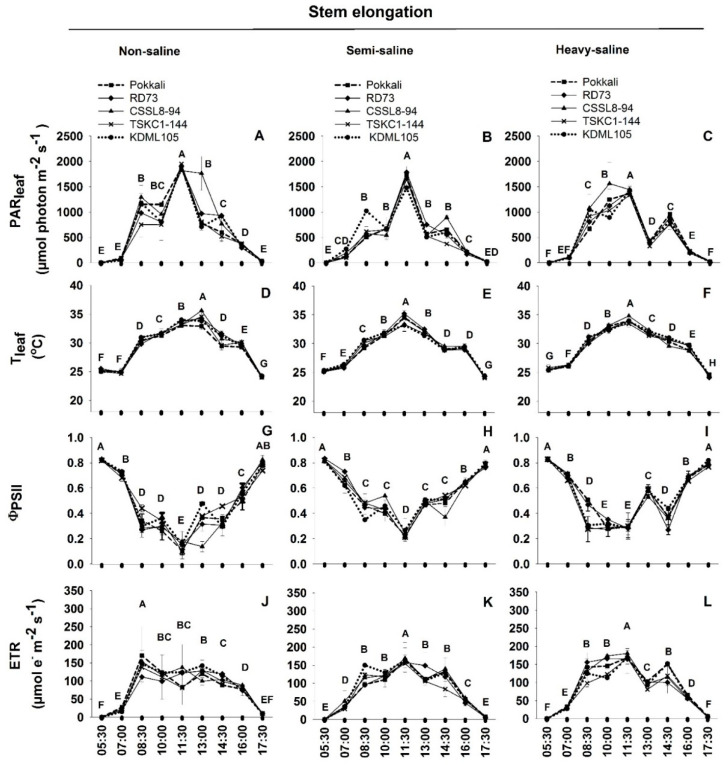
The variation in diurnal patterns in physical parameters including PAR_leaf_ (**A**–**C**) and T_leaf_ (**D**–**F**) and chlorophyll fluorescence parameters including effective quantum yield of PSII photochemistry (ΦPSII, (**G**–**I**)) and electron transport rate (ETR, (**J**–**L**)) of five rice genotypes (■, Pokkali; ◆, RD73; ▲, CSSL8–94; **X**, TSKC1–144; and ●, KDML105) at the stem elongation stage. Rice plants were grown in the rice-growing season of 2021 under non-saline, semi-saline, and heavy-saline field conditions. The measurements were conducted from 05:30 to 17:30 on 16 September 2021. Each capital letter at each time point represents the mean of five genotypes. Different capital letters indicated significant (*p* < 0.05) differences among different time points.

**Figure 5 plants-12-01903-f005:**
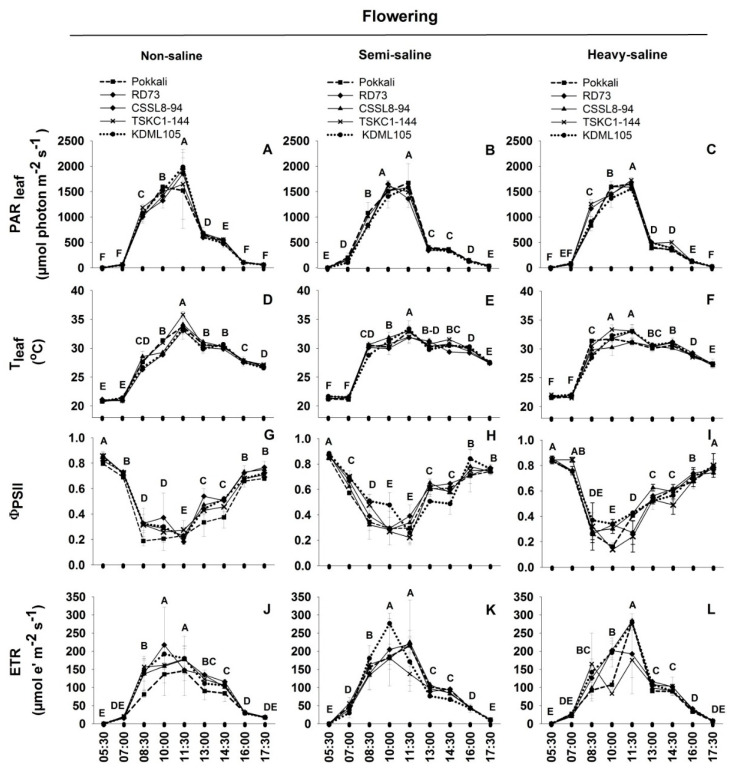
The variation in diurnal patterns in physical parameters including PAR_leaf_ (**A**–**C**) and T_leaf_ (**D**–**F**) and chlorophyll fluorescence parameters including effective quantum yield of PSII photochemistry (ΦPSII, (**G**–**I**)), and electron transport rate (ETR, (**J**–**L**)) of five rice genotypes (■, Pokkali; ◆, RD73; ▲, CSSL8–94; **X**, TSKC1–144; and ●, KDML105) at the flowering stage. Rice plants were grown in the rice-growing season of 2021 under non-saline, semi-saline, and heavy-saline field conditions. The measurements were conducted from 05:30 to 17:30 on 24 October 2021. Each capital letter at each time point represents the mean of five genotypes. Different capital letters indicated significant (*p* < 0.05) differences among different time points.

**Figure 6 plants-12-01903-f006:**
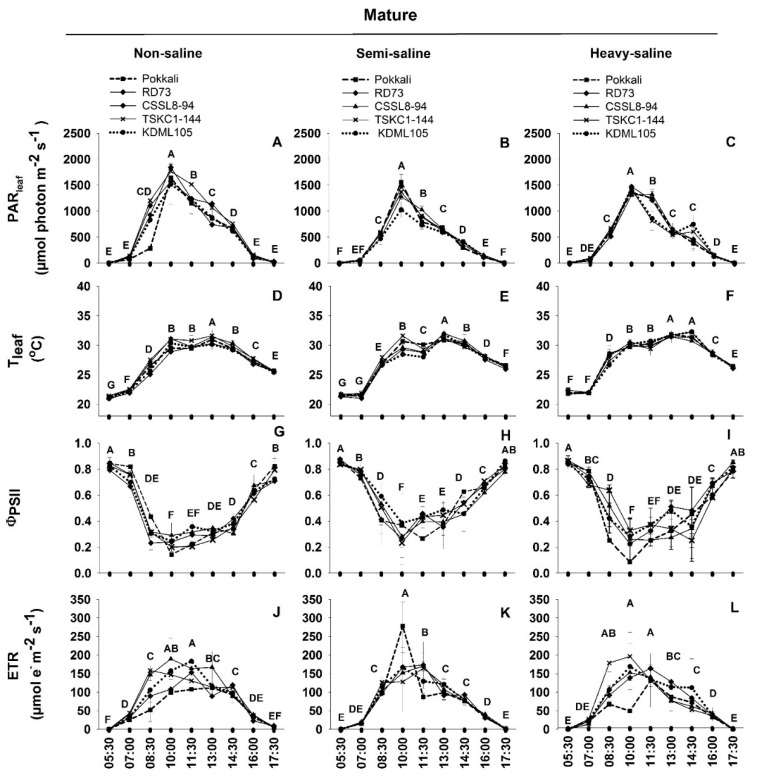
The variation in diurnal patterns in physical parameters including PAR_leaf_ (**A**–**C**) and T_leaf_ (**D**–**F**) and chlorophyll fluorescence parameters including effective quantum yield of PSII photochemistry (ΦPSII, (**G**–**I**)) and electron transport rate (ETR, (**J**–**L**)) of five rice genotypes (■, Pokkali; ◆, RD73; ▲, CSSL8–94; **X**, TSKC1–144; and ●, KDML105) at the mature stage. Rice plants were grown in the rice-growing season of 2021 under non-saline, semi-saline, and heavy-saline field conditions. The measurements were conducted from 05:30 to 17:30 on 9 November 2021. Each capital letter at each time point represents the mean of five genotypes. Different capital letters indicated significant (*p* < 0.05) differences among different time points.

**Figure 7 plants-12-01903-f007:**
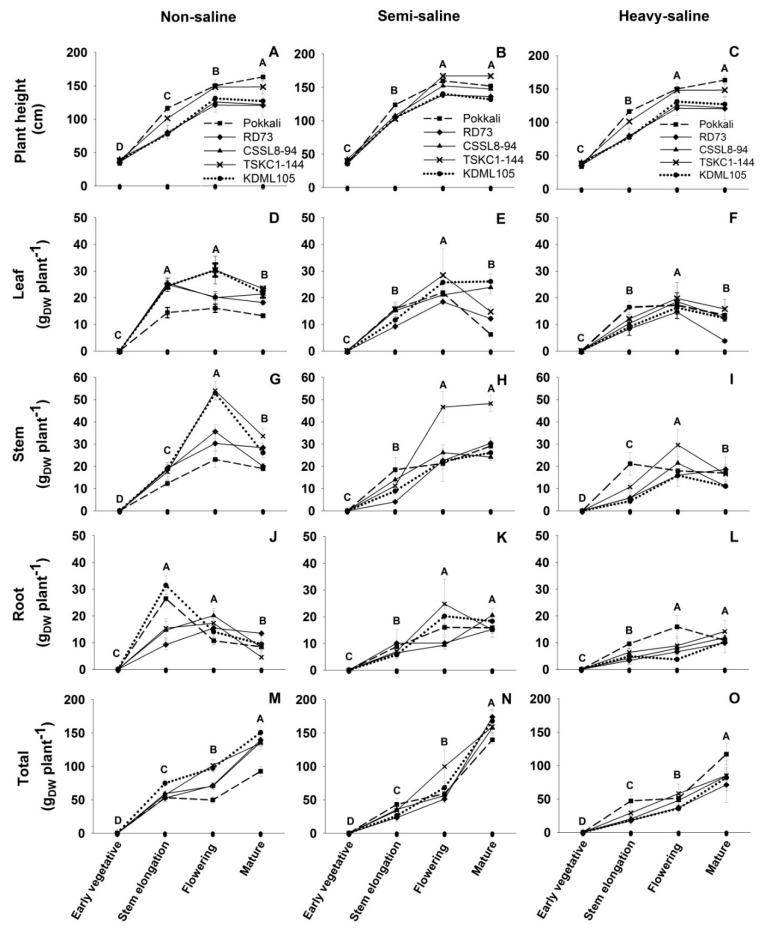
Plant height (**A**–**C**) and biomass of leaf (**D**–**F**), stem (**G**–**I**), root (**J**–**L**), and whole plant (**M**–**O**) of five rice genotypes (■, Pokkali; ◆, RD73; ▲, CSSL8–94; **X**, TSKC1–144; and ●, KDML105) at four growth stages including early vegetative, stem elongation, flowering, and mature. Rice plants were grown in the rice-growing season of 2021 under non-saline, semi-saline, and heavy-saline field conditions. Each capital letter at each growth stage represents the mean of five genotypes. Different capital letters indicated significant (*p* < 0.05) differences among different growth stages.

**Figure 8 plants-12-01903-f008:**
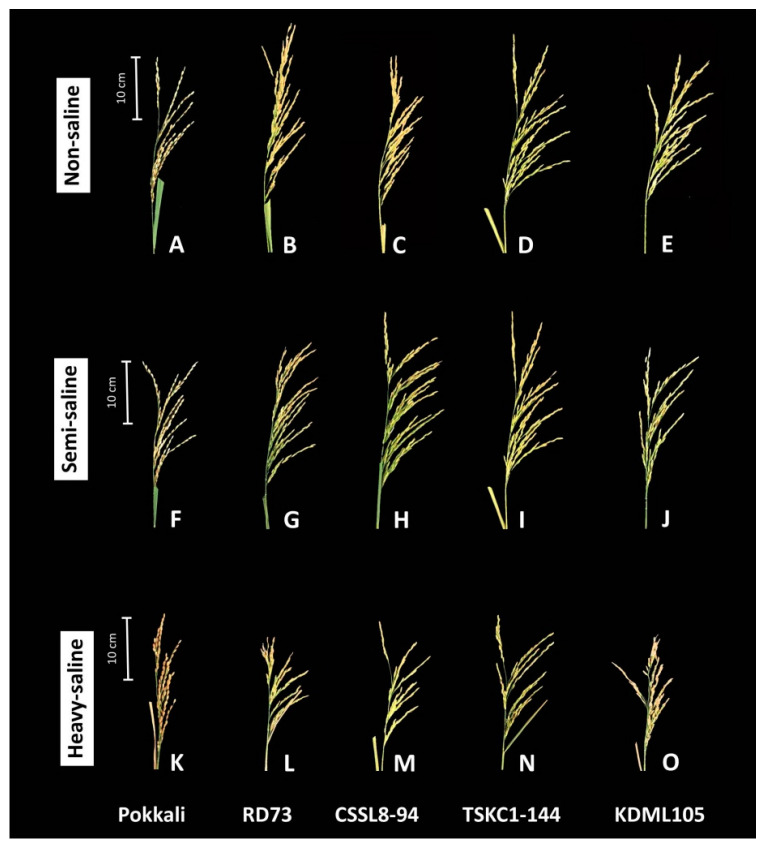
Panicles of five rice genotypes (Pokkali; RD73; CSSL8-94; TSKC1-144 and KDML105) at the mature stage. Rice plants were grown in the rice-growing season of 2021 under non-saline (**A**–**E**), semi-saline (**F**–**J**), and heavy-saline (**K**–**O**) field conditions.

**Figure 9 plants-12-01903-f009:**
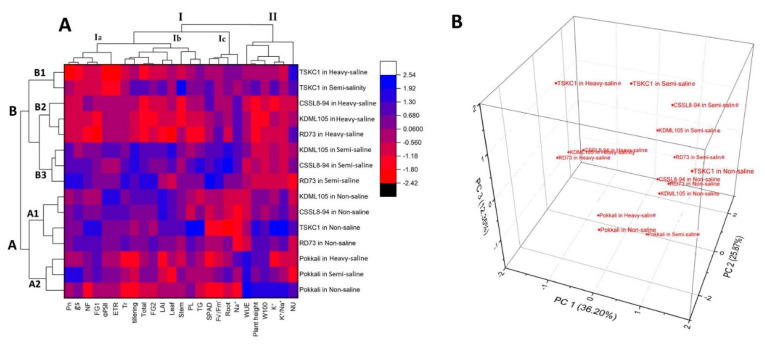
Hierarchical cluster analysis and heatmap explaining the responses of five rice genotypes (Pokkali, RD73, CSSL8–94, TSKC1–144 and KDML105) growing under different saline fields at mature stage (**A**). Columns correspond to dependent variables (biomass, yield, and physiological parameters), whereas rows correspond to different treatments (genotypes under different saline fields). Low numerical values are red, while high numerical values are blue (see the scale at the right corner of the heat map; P_n_ = net photosynthesis rate; gs = stomatal conductance; NF = number of filled grains; FG1 = filled grains weight panicle^−1^; PSII = effective quantum yield of PSII photochemistry; ETR = electron transport rate; T_r_ = transpiration rate; tillering = tiller number; total = total biomass; FG2 = filled grain weight plant^−1^; LAI = leaf area index; stem = stem biomass; PL = panicle length; TG = total grain number panicle^−1^; SPAD = leaf chlorophyll content; Fv′/Fm′ = maximum quantum yield of PSII photochemistry in the light; root = root biomass; Na^+^ = sodium ion content; WUE = water use efficiency; W100 = weight of 100 filled grains; K^+^ = potassium ion content; K^+^/Na^+^ = potassium to sodium ion ratio; NU = number of unfilled grains panicle^−1^). The principal component analysis (PCA) indicates that the rice genotypes based on photosynthesis, growth, and yield parameters recorded from the non-saline, semi-saline, and heavy-saline conditions at the mature stage (**B**).

**Table 1 plants-12-01903-t001:** Three-way ANOVA showing levels of significant differences for photosynthesis parameters of five rice genotypes growing under three different saline fields at six different growth stages.

Parameter	Field			Genotypes			Stage			Field × Genotypes			Field × Stage			Genotypes × Stage			Field × Genotypes × Stage		
	*df*	F	*p*	*df*	F	*p*	*df*	F	*p*	*df*	F	*p*	*df*	F	*p*	*df*	F	*p*	*df*	F	*p*
SPAD	2	185.2	<0.001	4	16.89	<0.001	5	85.88	<0.001	8	1.96	0.051	10	7.08	<0.001	20	4.16	<0.001	40	2.00	<0.001
P_n_ (µmol CO_2_ m^−2^ s^−1^)	2	1.14	0.321	4	8.385	<0.001	5	232.0	<0.001	8	4.20	<0.001	10	17.44	<0.001	20	1.81	0.019	40	2.00	<0.001
gs (mol H_2_O m^−2^ s^−1^)	2	21.41	<0.001	4	13.44	<0.001	5	160.9	<0.001	8	3.69	<0.001	10	73.60	<0.001	20	9.12	<0.001	40	1.97	<0.001
T_r_ (mmol H_2_O m^−2^ s^−1^)	2	79.37	<0.001	4	21.33	<0.001	5	29.43	<0.001	8	1.83	0.072	10	69.83	<0.001	20	3.46	<0.001	40	2.04	<0.001
WUE (µmol CO_2_ mmol H_2_O^−1^)	2	96.92	<0.001	4	15.09	<0.001	5	115.2	<0.001	8	6.53	<0.001	10	75.88	<0.001	20	9.60	<0.001	40	2.60	<0.001
ΦPSII	2	11.38	<0.001	4	1.734	0.143	5	182.3	<0.001	8	1.52	0.151	10	9.99	<0.001	20	1.91	0.012	40	1.78	0.004
Fv’/Fm’	2	81.39	<0.001	4	34.5	<0.001	5	140.6	<0.001	8	1.45	0.177	10	3.29	<0.001	20	3.62	<0.001	40	1.24	0.167
ETR (µmol e^−^ m^−2^ s^−1^)	2	11.39	<0.001	4	1.717	0.146	5	182.2	<0.001	8	1.53	0.148	10	9.99	<0.001	20	1.91	0.012	40	1.78	0.004

**Table 2 plants-12-01903-t002:** Three-way ANOVA showing levels of significance for different growth and biomass parameters of five rice genotypes growing under three different saline fields at four different growth stages.

Parameter	Field			Genotypes			Stage			Field × Genotypes			Field × Stage			Genotypes × Stage			Field × Genotypes × Stage		
	*df*	F	*p*	*df*	F	*p*	*df*	F	*p*	*df*	F	*p*	*df*	F	*p*	*df*	F	*p*	*df*	F	*p*
Plant height (cm)	2	69.44	<0.001	4	35.89	<0.001	3	1605	<0.001	8	2.28	0.024	6	8.94	<0.001	12	6.27	<0.001	24	1.12	0.325
Leaf dry weight (g_DW_ plant^−1^)	2	37.81	<0.001	4	9.63	<0.001	3	231	<0.001	8	3.77	<0.001	6	7.43	<0.001	12	3.18	<0.001	24	1.93	0.009
Stem dry weight (g_DW_ plant^−1^)	2	51.97	<0.001	4	15.04	<0.001	3	283	<0.001	8	5.98	<0.001	6	16.65	<0.001	12	7.44	<0.001	24	2.43	<0.001
Root dry weight (g_DW_ plant^−1^)	2	17.52	<0.001	4	1.723	0.147	3	86	<0.001	8	1.52	0.153	6	16.64	<0.001	12	2.05	0.022	24	2.86	<0.001
Total dry weight (g_DW_ plant^−1^)	2	60.01	<0.001	4	2.88	0.024	3	561	<0.001	8	5.08	<0.001	6	21.98	<0.001	12	2.62	0.003	24	1.37	0.128

**Table 3 plants-12-01903-t003:** Two-way ANOVA showing level of significance for characters at plant maturity including ion contents, and yield components of five rice genotypes growing under three saline fields.

Parameter	Field			Genotypes			Field × Genotypes		
	*df*	F	*p*	*df*	F	*p*	*df*	F	*p*
Na content (%)	2	121.59	<0.001	1	14.09	<0.001	2	8.39	<0.001
K content (%)	2	14.49	<0.001	1	0.51	0.478	2	1.08	0.343
K/Na ratio	2	149.11	<0.001	1	10.16	0.002	2	3.37	0.038
Panicle length (cm)	2	5.01	0.011	4	2.65	0.046	8	3.36	0.004
Total grain number panicle^−1^	2	2.19	0.124	4	1.92	0.124	8	1.02	0.438
No. of filled grains panicle^−1^	2	3.32	0.050	4	0.56	0.692	8	1.92	0.081
No. of unfilled grains panicle^−1^	2	0.54	0.589	4	6.07	<0.001	8	2.55	0.022
Filled grain weight panicle^−1^ (g)	2	5.84	0.006	4	0.71	0.588	8	2.36	0.033
Filled grain weight plant^−1^ (g)	2	36.29	<0.001	4	1.81	0.143	8	4.30	<0.001
Weight of 100 filled grains (g)	2	55.80	<0.001	4	51.16	<0.001	8	2.10	0.056

**Table 4 plants-12-01903-t004:** Sodium (Na^+^), Potassium (K^+^), and K^+^/Na^+^ ratio of five rice genotypes growing under non-saline, semi-saline, and heavy saline field conditions. Significant differences among genotypes in each column are indicated by different lower-case letters. Significant differences among the fields in each row are indicated by different capital letters.

Rice Genotype	Genotype	Saline Field			^2^ Critical-*p*
		Non-Saline	Semi-Saline	Heavy-Saline	Value
	Pokkali	0.006 ± 0.001 C	0.014 ± 0.006 cB	0.027 ± 0.003 A	*p* < 0.01
	RD73	0.007 ± 0.001 B	0.024 ± 0.006 bA	0.028 ± 0.006 A	*p* < 0.01
**Na content (%)**	CSSL8-94	0.007 ± 0.002 B	0.030 ± 0.001 aA	0.031 ± 0.002 A	*p* < 0.01
	TSKC1-144	0.007 ± 0.001 C	0.016 ± 0.005 cB	0.026 ± 0.008 A	*p* < 0.01
	KDML105	0.008 ± 0.003 B	0.030 ± 0.002 aA	0.030 ± 0.002 A	*p* < 0.01
	^1^ Critical-*p* value	ns	*p* < 0.01	ns	
	**mean**	**0.007 C**	**0.023 B**	**0.028 A**	*p* < 0.01
	Pokkali	0.852 ± 0.218 abA	0.683 ± 0.207 AB	0.472 ± 0.141 B	*p* < 0.01
	RD73	0.732 ± 0.124 bcA	0.546 ± 0.149 B	0.488 ± 0.187 B	*p* < 0.01
**K content (%)**	CSSL8-94	0.655 ± 0.095 cA	0.642 ± 0.100 A	0.432 ± 0.203 B	*p* < 0.01
	TSKC1-144	0.952 ± 0.115 aA	0.754 ± 0.320 B	0.602 ± 0.124 B	*p* < 0.01
	KDML105	0.650 ± 0.232 c	0.591 ± 0.109	0.592 ± 0.268	ns
	^1^ Critical-*p* value	*p* < 0.01	ns	ns	
	**mean**	**0.768 A**	**0.643 B**	**0.517 C**	*p* < 0.01
	Pokkali	123 ± 25 aA	52 ± 23 aB	16 ± 4 C	*p* < 0.01
	RD73	94 ± 20 bA	22 ± 4 bB	18 ± 5 B	*p* < 0.01
**K/Na ratio**	CSSL8-94	91 ± 25 bA	21 ± 4 bB	13 ± 6 B	*p* < 0.01
	TSKC1-144	138 ± 24 aA	45 ± 12 aB	24 ± 6 C	*p* < 0.01
	KDML105	81 ± 27 bA	19 ± 3 bB	19 ± 10 B	*p* < 0.01
	^1^ Critical-*p* value	*p* < 0.01	*p* < 0.01	ns	
	**mean**	**106 A**	**32 B**	**18 C**	*p* < 0.01

^1^ Critical-*p* value for testing each trait among rice genotype (in each column) and ^2^ critical-*p* value for testing each trait among saline fields (in each row).

**Table 5 plants-12-01903-t005:** Panicle length, total grain number panicle^−^^1^, number of filled and unfilled grains panicle^−^^1^, filled grain weight panicle^−^^1^, filled grain weight plant**^−1^**, and weight of 100 filled grains of rice growing under non-saline, semi-saline, and heavy-saline field conditions.

Trait	Genotype	Saline Field			^1^ Critical-*p*
		Non-Saline	Semi-Saline	Heavy-Saline	Value
	Pokkali	28.9 ± 1.6 bc	27.5 ± 0.9 (−5%)	28.5 ± 1.1 (−1%)	ns
	RD73	30.7 ± 1.3 abA	28.8 ± 1.1 AB (−6%)	24.1 ± 1.7 B (−21%)	*p* < 0.01
**Panicle length**	CSSL8-94	25.4 ± 2.1 c	29.9 ± 0.7 (+18%)	27.9 ± 0.3 (+10%)	ns
**(cm)**	TSKC1-144	33.7 ± 0.8 aA	29.6 ± 1.3 AB (−12%)	26.9 ± 1.4 B (−20%)	*p* < 0.01
	KDML105	26.9 ± 0.7 bc	27.3 ± 0.4 (+1%)	26.3 ± 1.3 (−2%)	ns
	^2^ Critical-*p* value	*p* < 0.05	ns	ns	
	**mean**	**29.1 ± 1.6 A**	**28.6 ± 0.6 A**	**26.7 ± 0.8 B**	** *p* ** **< 0.05**
	Pokkali	162 ± 20	170 ± 17 (+5%)	165 ± 10 (+2%)	ns
	RD73	181 ± 21	184 ± 16 (+2%)	151 ± 20 (−17%)	ns
**Total grain**	CSSL8-94	183 ± 22	213 ± 15 (+14%)	166 ± 5 (−1%)	ns
**number/panicle**	TSKC1-144	233 ± 27	181 ± 24 (−23%)	181 ± 27 (−36%)	ns
	KDML105	152 ± 4	185 ± 22 (+22%)	154 ± 11 (−6%)	ns
	^2^ Critical-*p* value	ns	ns	ns	
	**mean**	**182 ± 15**	**186 ± 7**	**163 ± 7**	**ns**
	Pokkali	106 ± 12 c	157 ± 17 (+48%)	143 ± 9 (+35%)	ns
	RD73	163 ± 19 ab	172 ± 17 (+6%)	101 ± 34 (−38%)	ns
**No. of filled**	CSSL8-94	146 ± 12 ab	167 ± 6 (+14%)	145 ± 1 (−1%)	ns
**grains/panicle**	TSKC1-144	175 ± 14 a	119 ± 25 (−32%)	112 ± 38 (−36%)	ns
	KDML105	127 ± 2 bc	155 ± 17 (+22%)	119 ± 16 (−6%)	ns
	^2^ Critical-*p* value	*p* < 0.05	ns	ns	
	**mean**	**143 ± 13**	**154 ± 10**	**124 ± 9**	**ns**
	Pokkali	55 ± 15 A	13 ± 1 bcB (−76%)	15 ± 4 cB (−73%)	*p* < 0.01
	RD73	18 ± 3 B	12 ± 2 cB (−33%)	50 ± 15 abA (+178%)	*p* < 0.01
**No. of unfilled**	CSSL8-94	37 ± 12	46 ± 11 ab (+24%)	20 ± 4 c (−46%)	ns
**grains/panicle**	TSKC1-144	58 ± 14	61 ± 20 a (+5%)	69 ± 13 a (+19%)	ns
	KDML105	24 ± 5	30 ± 11 a–c (+25%)	35 ± 5 bc (+46%)	ns
	^2^ Critical-*p* value	ns	*p* < 0.05	*p* < 0.05	
	**mean**	**38 ± 9**	**32 ± 10**	**38 ± 11**	ns
	Pokkali	3.42 ± 0.38 c	4.77 ± 0.54 (+39%)	4.42 ± 0.31 (+29%)	ns
	RD73	4.70 ± 0.53 abA	4.38 ± 0.52 A (−7%)	2.22 ± 0.79 B (−53%)	*p* < 0.01
**Filled grain**	CSSL8-94	3.98 ± 0.29 bc	4.28 ± 0.11 (+8%)	3.58 ± 0.03 (−10%)	ns
**weight/panicle**	TSKC1-144	5.08 ± 0.21 a	3.38 ± 0.75 (−33%)	3.03 ± 1.08 (−40%)	ns
(g)	KDML105	3.76 ± 0.07 bc	4.00 ± 0.38 (+6%)	2.80 ± 0.50 (−26%)	ns
	^2^ Critical-*p* value	*p* < 0.05	ns	ns	
	**mean**	**4.19 ± 0.34 A**	**4.16 ± 0.26 A**	**3.21 ± 0.42 B**	***p* < 0.05**
	Pokkali	14.98 ± 1.78 c	24.62 ± 1.59 (+64%)	21.27 ± 3.33 a (+42%)	ns
	RD73	25.27 ± 1.19 aB	31.26 ± 1.94 A (+24%)	13.95 ± 3.42 bC (−45%)	*p* < 0.01
**Filled grain**	CSSL8-94	21.73 ± 1.08 abB	24.58 ± 1.33 A (+13%)	14.07 ± 1.79 bB (−35%)	*p* < 0.01
**weight/plant**	TSKC1-144	21.10 ± 1.31 b	22.41 ± 3.20 (+6%)	14.41 ± 1.29 b (−32%)	ns
(g)	KDML105	25.36 ± 1.02 aA	26.46 ± 2.80 A (+4%)	10.77 ± 1.04 bB (−58%)	*p* < 0.01
	^2^ Critical-*p* value	*p* < 0.01	ns	*p* < 0.05	
	**mean**	**21.68 ± 2.12 B**	**25.86 ± 1.67 A**	**14.89 ± 1.93 C**	** *p* ** **< 0.01**
	Pokkali	3.39 ± 0.06 aA	3.18 ± 0.02 aB (−6%)	3.17 ± 0.04 aB (−6%)	*p* < 0.05
	RD73	3.02 ± 0.04 bA	2.58 ± 0.07 bB (−15%)	2.40 ± 0.07 cB (−21%)	*p* < 0.05
**Weight**	CSSL8-94	2.70 ± 0.06 c	2.59 ± 0.06 b (−4%)	2.41 ± 0.09 bc (−11%)	ns
**of 100 filled** **grains**	TSKC1-144	3.09 ± 0.09 b	2.71 ± 0.11 b (−12%)	2.62 ± 0.08 b (−15%)	ns
(g)	KDML105	2.93 ± 0.03 b	2.51 ± 0.08 b (−14%)	2.38 ± 0.01 c (−19%)	*p* < 0.05
	^2^ Critical-*p* value	*p* < 0.05	*p* < 0.05	*p* < 0.05	
	**mean**	**3.03 ± 0.12 A**	**2.71 ± 0.13 B**	**2.59 ± 0.17 C**	** *p* ** **< 0.05**

Means which are significantly different (*p* < 0.05 and *p* < 0.01) among genotypes for each saline field are denoted by different lower-case letters. Capital letters indicate significant (*p* < 0.05 and *p* < 0.01) differences of means among saline fields. Data show mean of four replicates ± standard error (SE). ^1^ Critical-*p* value for testing each trait among rice genotype (in each column) and ^2^ critical-*p* value for testing each trait among saline fields (in each row).

**Table 6 plants-12-01903-t006:** The soil physicochemical properties at pre-planting in early rainy season (May, 2021) in experimental fields including non-saline, semi-saline, and non-saline at soil depths of 0–15 cm.

Soil Physicochemical Properties	Non-Saline	Semi-Salinity	Heavy-Salinity
Physical properties			
Soil Texture	Sandy Loam	Loam	Loam
Sand (%)	71.82	40.75	37.33
Silt (%)	23.44	33.67	37.56
Clay (%)	4.74	25.58	25.11
Chemical properties at pre-planting			
Total N (%)	0.031 ± 0.001	0.083 ± 0.001	0.041 ± 0.001
Available P (mg kg^−1^)	46.15 ± 0.81	15.56 ± 0.92	9.23 ± 0.44
Exchangeable K (mg kg^−1^)	31.96 ± 0.86	69.48 ± 1.25	52.68 ± 1.45
pH (1:1 H_2_O)	6.67 ± 0.01	7.176 ± 0.02	7.113 ± 0.01
EC (dS m^−1^)	0.12 ± 0.01	2.87 ± 0.02	2.53 ± 0.01
SOM (%)	0.59 ± 0.03	1.21 ± 0.04	0.44 ± 0.11

N: nitrogen; P: phosphorus; K: potassium; EC: electrical conductivity; SOM: soil organic matter. Values are mean ± standard error of four replications.

## Data Availability

Data are contained within the article.

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
