# Peer review of "Photosynthesis Performance at Different Growth Stages, Growth, and Yield of Rice in Saline Fields"

_plants, 2023, doi:10.3390/plants12091903_

Round 1
Reviewer 1 Report (Previous Reviewer 1)
The authors carried out a detailed study on the effect of salinity on rice photosynthesis and growth. That provides some advanced information on the salt stress response in their local rice cultivars. However, the presentation of the results is complex and could be improved for readability. Thanks.
Author Response
Comments and Suggestions for Authors
The authors carried out a detailed study on the effect of salinity on rice photosynthesis and growth. That provides some advanced information on the salt stress response in their local rice cultivars. However, the presentation of the results is complex and could be improved for readability. Thanks.
Author response: Thank you very much for your comments. We have now improved the presentation of results as follows: (1) We now present the ANOVA Tables for overall information on the levels of significance of each factor i.e., rice genotypes, growth stages, saline fields, and their interactions, (2) For comparisons between saline and non-saline field, we now added percentage changes of some important parameters (such as photosynthesis rates, total biomass, and yield characters) in the brackets following the mean values in Table 5, Supplement Table S2 and S3, (3) We performed PCA analysis using data of all physiological and agronomic parameters at the mature stage, to explain similarity in responses and clustering of genotypes in response to different salinity levels. The PCA and HCA results are displayed in Figure 9.

Reviewer 2 Report (New Reviewer)
Deat Authors,
the article you prepared is very well written and organized. Everything is understandable and nicely explained. I have few minor comments that I placed directly in the uploaded document.
Best regards.

Author Response
Response to Reviewer 2
Comments and Suggestions for Authors
Dear Authors, the article you prepared is very well written and organized. Everything is understandable and nicely explained. I have few minor comments that I placed directly in the uploaded document.
Author response: Thank you so much for your kind advice. We have corrected all points you suggested in your pdf file as listed below:
- It is custom to use expression "meteorological", not "environmental"
Author Response: We replaced ‘environmental’ with ‘meteorological’
- I suggest to use uniform legend position in all figures and it is easier to read when the legend is above the figure (as is in Figure 4).
Author response: Figure legend must be below the figure according to the Jounal’s format.
- Line 1102: There is no plural of rainfall, so no rainfalls, just rainfall
Author response: We removed ‘s’.
- Line 1102: ‘annual rainy days was 111 days’ Average number or?
Author response: total number of rainfall days in one year.
- Line 1104
Author response: Already changed ‘environmental’ to ‘meteorological’
- Line 1111 – 1113 - This sentence is not clear and there are several mistakes that need to be corrected.
Author response: this sentence has been rewritten as follows: - ‘The non-saline plot was located approximately 300 m from the non-saline one but on a higher ground of about 3 m.’

Reviewer 3 Report (New Reviewer)
Nice job done to a high standard. I think that it can be published as is or with a few comments below.
The analysis of the results is quite sufficient for this stage of the work. However, the nature of the data obtained by the authors suggests that, with a deeper analysis, additional information on the mutual influence of photosynthesis and productivity parameters can be obtained, for example, when using the principal component method and related statistical methods. I hope that the authors will conduct such an analysis in the future and publish its results.
K195 Please decipher the QTL on first use.
L213-214 1 decimal place is sufficient to represent temperature and humidity data.
Author Response
Response to Reviewer 3
Comments and Suggestions for Authors
Nice job done to a high standard. I think that it can be published as is or with a few comments below.
The analysis of the results is quite sufficient for this stage of the work. However, the nature of the data obtained by the authors suggests that, with a deeper analysis, additional information on the mutual influence of photosynthesis and productivity parameters can be obtained, for example, when using the principal component method and related statistical methods. I hope that the authors will conduct such an analysis in the future and publish its results.
Author response: We are grateful for your kind suggestions. In this revised version we have conducted PCA and displayed the results in Figure 9.
K195 Please decipher the QTL on first use.
Author response: We have added ‘quantitative trait loci (QTL)’
L213-214 1 decimal place is sufficient to represent temperature and humidity data.
Author response: Temperature and humidity data were corrected to one decimal place.

Reviewer 4 Report (New Reviewer)
The manuscript addresses an interesting and actual worldwide issue: the effect of salt stress on rice plants at different growth stages and saline conditions. The manuscript has a good experimental design behind it. The experimental design is well planned: the analyses conducted could provide a comprehensive overview for answering the questions posed by the authors. On the contrary, statistical methods are not complete and correctly applied (please check the file attached). In addition, some data were reported and discussed as mean across genotypes and growth stages, unfortunately these data were misleading because each genotype/growth stage has a different trend, especially in salt stress conditions. So the obtained data should be analyzed and discussed by considering these factors separately and they should be confirmed through appropriate statistics. For these reasons, I was not able to revision properly Results and Discussions paragraphs.
It seems that the study is not new, as there are already studies on the response of these improved rice lines against salt stress. I would suggest emphasizing the novelty of this study under different point of view (e.g., pre-transplanting flooding or different environmental conditions at the site, if not already seen in other works).
Considering the well-planned of the study, I’ll encourage the authors to resubmit the manuscript with reported suggestions.
Please find attached the revisions of the manuscript.

Author Response
We are grateful for your valuable comments.
Please see attached file for our response.

Reviewer 5 Report (New Reviewer)
Dear authors,
This is an interesting and well written manuscript.
I do not have any specific comment to do or error to notice.
Author Response
Comments and Suggestions for Authors
Dear authors,
This is an interesting and well written manuscript.
I do not have any specific comment to do or error to notice.
Author response: We are grateful for your kind compliment and valuable support.

Round 2
Reviewer 4 Report (New Reviewer)
The changes made by the authors cleared up any doubts and explained the less clear points. I am glad that the appropriate statistical analysis did not distort the results of the work.
My only suggestion is to move Tables 1 and 3 to the supplementary material.
Thank you.
This manuscript is a resubmission of an earlier submission. The following is a list of the peer review reports and author responses from that submission.
Round 1
Reviewer 1 Report
This article indicates salinity stress is a constraint for rice planting in Thailand. The authors compared a salt-tolerant, a salt sensitive, and four improved cultivars, for the growth and photosynthesis performance in the fields with 3 levels of salinity stresses. This study improves our knowledge about rice responds to salt stress in the fields across different growth stages. Unfortunately, they just showed biomass data but not grain yield data, which is the most crucial trait in terms of economics. As the authors mentioned, “rice is most sensitive to salt stress at the reproductive” (Line 386).
1. Suggest to change ‘leaf greenness’ to “leaf chlorophyll content” in all text.
2. Line 166, normally, salt stress should induce stomatal closure and cause low gs. But this study showed the opposite phenomenon, need to discuss why?
3. Line 226, Different capital letters indicated that the mean across “genotypes” was significantly different… Should be “across different growth stages”?
4. Line 409, “Pokkali under the heavy-saline condition, despite the huge reduction in gs (Figure 2I), could be attributed to the protection of chloroplast structure and function as a result of the good maintenance of leaf water status following significantly lower Tr (Figure 2L) and prominently high WUE”. It is rare Pokkali increase photosynthesis under salt stress. Whether it has been reported before? Authors should discuss more the underlying mechanism of salt tolerance of Pokkali or cite others’ findings and discuss. SKC1 (HKT1) in Pokkali?
5. Whether SKC1 (salt tolerance gene on chromosome 1) is QTL from Salto? Any improvement under salt stress?
6. Line 566, “KDML105 genetic background resulting from a cross be-566 tween CSSL8-94 and RGD4 (KDML105 introgressed with SKC1), therefore SKC1 and DT-567 QTL8 were pyramided in this line [55].”
7. Please show the calculation method of WUE in M&M.
8. Line 646, “ETR value was calculated from the equation: ETR = ФPSII x 0.84 x 0.5 x PAR”. I think it is no need to mention it here. Li6400 has ETR data.
9. Minor changes were noted in the pdf text.
10. English should be improved.

Author Response
This article indicates salinity stress is a constraint for rice planting in Thailand. The authors compared a salt-tolerant, a salt sensitive, and four improved cultivars, for the growth and photosynthesis performance in the fields with 3 levels of salinity stresses. This study improves our knowledge about rice responds to salt stress in the fields across different growth stages. Unfortunately, they just showed biomass data but not grain yield data, which is the most crucial trait in terms of economics. As the authors mentioned, “rice is most sensitive to salt stress at the reproductive” (Line 386).
Author response: We highly appreciate your constructive comments which greatly help improve our manuscript. In this paper we focused on reporting the photosynthetic performance of rice in the saline compared with the non-saline field, therefore we presented only the biomass data. Regarding your concern about the grain yield data, we plan to present them in another manuscript which will include both grain yield and quality. We thank you in advance for your understanding.
- Suggest to change ‘leaf greenness’ to “leaf chlorophyll content” in all text.
Author response: We have changed ‘leaf greenness’ to ‘leaf chlorophyll content’ throughout the manuscript
- Line 166, normally, salt stress should induce stomatal closure and cause low gs. But this study showed the opposite phenomenon, need to discuss why?
Author response: At early vegetative stage (Fig. 2G, H, I), the mean gs of the non-saline was lower (» 0.12 mmol H2O m-2 s-1) than the semi-saline and heavy saline fields (» 0.48 and 0.47 mmol H2O m-2 s-1) even though the measurements were taken on the same day during 9.00-11.30 am. The reason for this unusual phenomenon was because the measurements in the saline fields were done earlier (during 9.00 – 10.30) while those in the non-saline field were done later after 11.00 am when the air/leaf temperature and light intensity were much higher (As an example, please see Fig. 4A for diurnal light intensity and Fig.4D for diurnal leaf temperature). The reason for different periods of measurement was because (1) we have only one set of equipment, (2) the non-saline and the two saline fields were approximately 300 meters apart, so we needed at least 30 – 45 min for packing up the equipment after completing the work in the two saline fields, transport, and preparation of the equipment for working the non-saline field. We add the discussion regarding this in Lines 263 - 266
- Line 226, Different capital letters indicated that the mean across “genotypes” was significantly different… Should be “across different growth stages”?
Author response: Line 226 (in the original version) was the legend of Figure 3. We clarified this by changing the text to “Each capital letter at each growth stage represents the mean of five genotypes. Different capital letters indicated significant differences among different growth stages”. Likewise, we also changed the text in the legend of Figure 2. [For significant differences among different growth stages of each genotype, please see Supplementary Table S2]. Moreover, Table S2 also showed the mean across six growth stages of each genotype which can be compared among different genotypes.
- Line 409, “Pokkali under the heavy-saline condition, despite the huge reduction in gs (Figure 2I), could be attributed to the protection of chloroplast structure and function as a result of the good maintenance of leaf water status following significantly lower Tr (Figure 2L) and prominently high WUE”. It is rare Pokkali increase photosynthesis under salt stress. Whether it has been reported before? Authors should discuss more the underlying mechanism of salt tolerance of Pokkali or cite others’ findings and discuss. SKC1 (HKT1) in Pokkali?
Author response:
- “It is rare Pokkali increase photosynthesis under salt stress. Whether it has been reported before?”
We found only one publication, a pot experiment, showing that photosynthesis rate of Pokkali under heavy salinity (12 dS m-1) did not differ significantly from the control while the rate of CSR-13 (another salt-tolerant variety) under salt stress was significantly reduced. [Pal et al. 2004. Photosynthetic characteristics and activity of antioxidant enzymes in salinity tolerant and sensitive rice cultivars. Indian Journal of Plant Physiology 9(4): 407 – 412]. This report of Pal et al. 2004, highlighted that photosynthesis performance of Pokkali was resistant to high salinity. In our field experiment, photosynthesis ability of Pokkali was not reduced but was even higher in saline fields than non-saline field. It could be because (1) excellent root systems of Pokkali extended to areas where salinity was very low (salinity in soils in nature is not homogeneous, unlike the situation in the pots) (2) low salt concentration may stimulate growth of Pokkali similar to the beneficial effects of low salt concentration in halophytes. This photosynthetic characteristic of Pokkali interesting and needs to be studied further.
- Authors should discuss more the underlying mechanism of salt tolerance of Pokkali or cite others’ findings and discuss. SKC1 (HKT1) in Pokkali?
Author response: We have discussed the mechanism of SKC1 in Lines 774 - 779.
- Whether SKC1 (salt tolerance gene on chromosome 1) is QTL from Salto? Any improvement under salt stress?
Author response: (1) Yes, SKC1 is the salt-tolerant gene located in Saltol QTL [Ren ZH, et al. 2005. A rice quantitative trait locus for salt tolerance encodes a sodium transporter. Nat Genet. 37(10):1141-6.] (2) In a previous report by Suriya-aroonroj et al. (2018) in Thai Rice Research Journal Vol. 9 No. 2 pp. 25-50, RD 73 (the improved cultivar from KDML105 introgressed with SKC1 gene) showed higher salt tolerance level than KDML105 based on salt tolerance scores, percent survival, biomass, yield, and harvest index. In our study, we found that SPAD values and Fv’/Fm’ of RD73 were higher than KDML105 in all three fields (non-saline, semi-saline, and heavy-saline); while net photosynthesis rate (Pn) of RD73 was slightly higher than that of KDML105 in non-saline and semi-saline fields; stem and root biomass of RD73 in the heavy-saline field were higher than those of KDML105 (Please see Supplementary Table S2).
- Line 566, “KDML105 genetic background resulting from a cross between CSSL8-94 and RGD4 (KDML105 introgressed with SKC1), therefore SKC1 and DT-567 QTL8 were pyramided in this line [55].”
Author response: This line (TSKC1-144) was a BC2F2 line derived from a cross between CSSL8-94 [an improved line of KDML105 introgressed with drought-tolerant QTL on chromosome 8 (DT-QTL8)] x RGD4 (an improved line of KDML105 introgressed with SKC1 gene), therefore this TSCK1-144 line carried both DT-QTL and SKC1 gene. This is one of the improved lines from Ph.D. research performed in our research group (Reference #57 Pamuta, 2021)
- Please show the calculation method of WUE in M&M.
Author response: The calculation for WUE was added in Line 861
- Line 646, “ETR value was calculated from the equation: ETR = ФPSII x 0.84 x 0.5 x PAR”. I think it is no need to mention it here. Li6400 has ETR data.
Author response: ETR equation was removed.
- Minor changes were noted in the pdf text.
Author response: Many thanks for your thorough and detailed review; all comments from provided in your attached pdf have been corrected.
- English should be improved.
Grammatical errors and unclear sentences have been corrected/modified.

Reviewer 2 Report
The study of salt stress seems to be very important because the area of saline soils will constantly increase. In addition, rice is an important crop. Therefore, research on rice growth under stress conditions are important. However, the study presented by Santana et al. is very poor quality. The authors worked on already described lines whose resistance to salt stress is known. Authors performed just simple analysis. Author suggest that it is first field experiment about salt stress response in rice and its effect on plants growth, but it is no true. Discussion is very descriptive and adds nothing to the current state of knowledge.
Few other comments:
Things exhibited on figure 1, 5 A-F and 6 A-F it is no results. I just observation on monitoring of environment.
The statistics used by the authors and their description in the figure legend do not allow to understand the results. The bad graphics don't help either.
Why are there no pictures of plants in the supplement?
Results are described very chaotic
Description fact that the fluorescence of chlorophyll changes during the day is nothing new
Same about affecting growth by stress, where is novelty?
Author Response
The study of salt stress seems to be very important because the area of saline soils will constantly increase. In addition, rice is an important crop. Therefore, research on rice growth under stress conditions are important. However, the study presented by Santana et al. is very poor quality. The authors worked on already described lines whose resistance to salt stress is known.
Author response: In this paper we studied 5 genotypes (Pokkali – the salt tolerant standard genotype widely used as the donor for SKC1 salt tolerant gene in breeding for salt tolerance; KDML105 – the elite aromatic rice cultivar which is sensitive to salinity and drought; RD73 – an improved cultivar (new cultivar released by Rice Department Thailand in 2017) recommended for farmers in saline areas. This is an improved cultivar from KDML105 introgressed with SKC1 gene); CSSL8-94 – an improved line of KDML105 introgressed with drought-tolerant QTL on chromosome 8; and TSKC1-155 – an improved line with KDML105 genetic background introgressed with both drought-tolerant QTL and SKC1 gene). Although in the process of breeding, breeders have proven the salt tolerance level (based on salt injury scores, survival, and growth) none of these genotypes were fully characterized in terms of ‘physiology’, particularly photosynthesis (with the exception of a report, in greenhouse experiment, from our research group on comparing photosynthesis performance of CSSL8-94 with KDML105 by Pamuta et al. 2020). The data on physiology and photosynthesis of RD73 and TSKC1-144 were totally lacking. Photosynthesis activity is vital for plant growth, therefore knowledge in photosynthesis performance in the real life field conditions (in our case non-saline VS saline fields) should be useful for plant breeders, physiologists, and agronomists.
Authors performed just simple analysis.
Author response: Data analysis in this work followed the experimental design i.e., ANOVA in RCBD.
Author suggest that it is first field experiment about salt stress response in rice and its effect on plants growth, but it is no true.
Author response: We removed this sentence from the manuscript.
Discussion is very descriptive and adds nothing to the current state of knowledge.
Author response: This study showed that the improved line CSSL8-94 which resulted from the introgression of drought-tolerant genes (located on DT-QTL8) into KDML105 genome, the line RD73 (introgressed with SKC1 salt-tolerant gene) had better growth in the heavy-saline field than KDML105. Moreover, the improved line TSKC1-144 (which carried both drought- and salt-tolerant genes) showed even higher growth than CSSL8-94 and RD73. This data on the benefit of pyramiding the two genes is first reported in this study. These improved lines proved to be potential sources of breeding materials for rainfed lowland rice areas which are affected by salinity. This experiment was carried out in the neglected saline areas where farmers never used for rice growing. We showed that with good cultural practice and land management, certain rice genotypes can be grown in this area.
Few other comments:
Things exhibited on figure 1, 5 A-F and 6 A-F it is no results. I just observation on monitoring of environment.
Author response: Photosynthesis performance constantly varies greatly during development of rice depending on the environment; therefore, we presented the environment data (weather and soil/water salinity) during the rice growing period to help explaining changes in photosynthesis throughout the growing stages in Figure 1. Environment also changes every minute – hour during the day which is the reason why we performed diurnal changes in photosynthesis in terms of Chl fluorescence. For Fig. 5A-F (flowering stage) and Fig. 6A-F (mature stage), A to C presented diurnal light intensity (photosynthetically active radiation, PAR) intercepted on the leaf surface, while D to F presented diurnal changes in leaf temperature; both leaf PAR and leaf temperature are the major factors affecting photosynthesis particularly efficiency of PSII in converting light energy into chemical energy in the form of electron transport (and hence ATP and NADPH synthesis). Therefore, Fig. A to C, and D to F were needed to explain the diurnal changes in PSII efficiency (FPSII) in Fig. G to I, and electron transport rate (ETR) in Fig. J to L.
The statistics used by the authors and their description in the figure legend do not allow to understand the results. The bad graphics don't help either.
Author response: The statistics shown in the Supplementary, for example Table S2, for each photosynthesis parameter (SPAD, Pn, gs, Tr, and WUE) readers can compare the significant difference among growth stages of each cultivar by the different capital letters, and the significant difference among five cultivars at each growth stage by different lowercase letters. The mean of five cultivars is typed in bold letters below the data of the five cultivars; comparison among growth stages is indicated by bold capital letters, and comparison among saline fields is indicated by bold lowercase letters. We modified the figure legends to be more precise.
Why are there no pictures of plants in the supplement?
Author response: We now added photos of saline fields and rice plants in Supplementary Figure S2
Results are described very chaotic
Author response: We tried to clarify many parts in the ‘Results’ to make them easier to understand.
Description fact that the fluorescence of chlorophyll changes during the day is nothing new
Author response: Although the diurnal change in fluorescence during the day is well-known, this type of data on field-grown rice, particularly comparison between non-saline VS saline fields, is rather limited. Our results showed that maximum PSII efficiency of all genotypes was well maintained even in flag leaves of mature plants as indicated by the values of Fv/Fm > 0.8 at both predawn and in the evening. At mature stage, even though the electron transport rates were not much different between non-saline and heavy-saline plants, the net photosynthesis rates in heavy-saline field were significant lower than the non-saline field. From this information, it is suggested that future breeding targets for improvement of photosynthesis performance under salinity should be more focused to the processes related to CO2 diffusion and assimilation.
Same about affecting growth by stress, where is novelty?
Author response: Our experiment was performed in saline fields where the farmers never used for growing rice because the presence of thick salt crust over the soil surface discouraged them from growing rice in this area. Our results showed that rice can be grown in this heavy saline field if the cultural practice is properly managed (See Supplementary Figure S2). Moreover, the results showed that the improved lines (CSSL8-94 and TSKC1-144) had higher total biomass than KDML105 in the heavy-saline field. This information confirmed that these two improved lines have good potential to be used as genetic resources for further rice breeding or to be promoted for commercial growing by the farmers.
Reviewer 3 Report
REVIEW REPORT
Ms title- Photosynthesis Performance at Different Growth Stages of Rice in Saline Fields
Journal: Plants
Authored by: Supranee Santanoo 1, Watanachai Lontom 1, Anoma Dongsansuk 2, Kochaphan Vongcharoen 3 and Piyada Theerakulpisut 1, *
1 Department of Biology, Faculty of Science, Khon Kaen University, Khon Kaen 40002, Thailand; supranee4705@hotmail.com (S.S.); watalo@kku.ac.th (W.L.)
2 Department of Agronomy, Faculty of Agriculture, Khon Kaen University, Khon Kaen 40002, Thailand; da- noma@kku.ac.th (A.D.)
3 Faculty of Science and Health Technology, Kalasin University, Kalasin 46000, Thailand; ko-cha_9@hot- 10 mail.com (K.V.)
Reviewer’s Recommendation: Major Revision
Reviewer’s comments to Authors-
1. Grammatical errors are present, please revise the whole manuscript to remove any possible grammatical errors, redundancy and typos.
2. The title of present study is fine however, the paper somewhat failed to answer some objectives of the study. This needs to be clearly addressed in the manuscript. Please revise it.
3. Error in sentence formation, please revise the whole manuscript to avoid the use of long sentences and some paragraphs are very short. Paraphrasing is also required in the manuscript.
4. Please maintain the uniformity while in-text citation and referencing.
5. In the keywords, it is strongly advisable use suitable words that can aid in finding out the manuscript in current registers or indexes. Strictly avoid the use of title words in the keywords.
6. The beginning of a new paragraph should be after some space and avoid beginning of new sentences with abbreviation, check in complete manuscript.
7. Uniformity in referencing is missing from the manuscript. Revise it.
8. It is recommended that authors should remove the below listed irrelevant self-cited papers from the manuscript-
· Santanoo, S.; Vongcharoen, K.; Banterng, P.; Vorasoot, N.; Joyloy, S.; Roytrakul, S.; Theerakulpisult, P. Seasonal variation in diurnal photosynthesis and chlorophyll fluorescence of four genotypes of cassava (Manihot esculenta Crantz) under irrigation conditions in a tropical savanna climate. Agronomy 2019, 9, 206.
· Pamuta, D.; Siangliw, M.; Sanitchon, J.; Pengrat, J.; Siangliw, J.L.; Toojinda, T.; Theerakulpisut, P. Photosynthetic performance in improved ‘KDML105’ rice (Oryza sativa L.) lines containing drought and salt tolerance genes under drought and salt stress. Pertanika J. Trop. Agric. Sci. 2020, 43, 653–675.
· Kanawapee, N.; Sanitchon, J.; Srihaban, P.; Theerakulpisut, P. Genetic diversity analysis of rice cultivars (Oryza sativa L.) differing in salinity tolerance based on RAPD and SSR markers. Electron. J. Biotechnol. 2011, 14, 1–17.
· Nounjan, N.; Wuttipong Mahakham, W.; Siangliw, J.L.; Toojinda, T.; Theerakulpisut, P. Chlorophyll retention and high photosynthetic performance contribute to salinity tolerance in rice carrying drought tolerance quantitative trait Loci (QTLs). Agriculture 2020, 10, 620.
Highlights, Abstract and Introduction:
1. Highlights section should be added; significant results must be included in the highlight section of this manuscript, adding a graphical abstract would also be helpful for summarizing the importance of this study.
2. Novelty statement is also omitted from the manuscript which is important for emphasizing the exclusivity of your study.
3. Add significant results from all the sections precisely in the abstract.
4. ‘‘Leaf greenness (SPAD values), on the other hand, remained high from the early vegetative to milky stage then reduced at maturity for all three conditions.” What does ‘for all three conditions’ signifies here, what conditions are being discussed is not clear in this line, please clarify.
5. “Rice response to salt stress is complex and depends on the type of salt stress, duration of salt exposue, and growth stage”, rephrase the sentence as it should be ‘’response of rice under salt stress….’, and correct the spelling of ‘exposure’.
6. “Accumulation of toxic Na+ in plant tissues disturbs metabolic processes, particularly photosynthesis, and all major morpho-physiological and yield-related traits including tiller number, panicle length, spikelet number per panicle”, toxic Na+ isn’t the correct term, either mention the toxic level for Na+ or rephrase the sentence.
7. “Immediate effects of salinity on photosynthesis of rice involved an induction of stomatal closure, due to salt-induced osmotic stress”, use the term “photosynthetic ability” instead of “photosynthesis”.
8. ‘‘This information will help determine genotypic potential and/or environmental factors limiting photosynthetic performance, and growth of rice. This information is expected to be useful for providing a set of guidelines for improving management of cultural practice for rice growing in saline fields.” This paragraph will be more suitable in the Conclusion section.
Materials and Methods:
1. ‘‘Seed germination was started on 18 June 2021, and seedlings were cultured at Khon Kaen Rice Research Center. When the plants were 40 days old, they were up--rooted and transplanted in the ploughed fields.’’ It is not clear whether the seed germination was done in pots or not, and how many replicates were planted, please specify.
2. ‘‘In early rainy season (May, 2021), soil EC of all plots were decreased from the hot season approximately 2.53, 2.87 and 0.12 for heavy-saline, semi-saline and non-saline plot, respectively’’. Does this constantly changing salinity of the soil due to varying weather conditions had any impact on the experiment, please explain.
Results and Discussion:
1. During the hot season (March to April), prior to the rice growing season, the mean total rainfall (114.1 mm). What does ‘mean total rainfall’ means? either it is total rainfall or it could be the mean of the total rainfall, please specify.
2. “Total rainfall, number of rainy days, monthly mean relative humidity (RH) and air temperature (T) of each month in 2021 were presented in Figure 1.” Why the complete ‘Result’ section is written in past tense when it should be in present tense, for example this sentence must be written as Total rainfall, number of rainy days, monthly mean relative humidity (RH) and air temperature (T) of each month in 2021 are presented in Figure 1. The use of ‘were’ should be replaced with ‘are’.
3. “The measurements at the flowering stage were conducted on flag leaves at the end of rainy season 21 days after water drainage from the rice fields and soil EC were highest.” How come the EC was highest after rainfall and water drainage as the water drainage after rainfall would result in the declining of salts present in the soil due to leaching, please explain.
4. “At maturity, the means plant height across genotypes in the non-, semi-, and heavy-saline fields were 166.2, 146.95, and 136.15 cm,”, this should be mean plant height, nor means.
5. The section of the results in which photosynthetic parameters are discussed is extended unnecessarily, this section should be rewritten.
6. Discussion section must more studies from previous reports as evidence for your study, recent studies must be included.
7. The authors failed to explain all of their findings in the discussion section with specific mechanisms. It is advisable to maintain balance between the result and the discussion section.
References:
There are lots of statements/sentences throughout the manuscript that essentially require proper validation and citation with previous studies which are largely missing in the present manuscript. Introduction, Result, and Discussion sections poorly cited with the references and strongly recommended to update and validation with previous studies. Therefore, some of the papers listed below should be considered and cited appropriately in the Introduction, Result, and Discussion sections of this manuscript which will certainly upgrade and enhance the Ms. quality significantly.
· Gupta, S., et al. (2021). Salicylic acid alleviates chromium (VI) toxicity by restricting its uptake, improving photosynthesis and augmenting antioxidant defense in Solanum lycopersicum L. Physiology and Molecular Biology of Plants, 27(11), 2651-2664.
· Yadav, M., et al. (2022). Foliar application of α-lipoic acid attenuates cadmium toxicity on photosynthetic pigments and nitrogen metabolism in Solanum lycopersicum L. Acta Physiologiae Plantarum, 44(11), 1-10.
· Gupta, P., et al. (2022). 24-Epibrassinolide Regulates Functional Components of Nitric Oxide Signalling and Antioxidant Defense Pathways to Alleviate Salinity Stress in Brassica juncea L. cv. Varuna. Journal of Plant Growth Regulation, 1-16.
· Prajapati, P., et al. (2022). Nitric oxide mediated regulation of ascorbate-glutathione pathway alleviates mitotic aberrations and DNA damage in Allium cepa L. under salinity stress. International Journal of Phytoremediation, 1-12.
Conclusion:
1. The conclusion section should be in accordance to the result and discussion section, random addition of results should be avoided.
2. Conclusion section failed to enlighten the spirit of the finding and is missing the results. Revise it precisely.
3. Conclusion section must also include the future perspectives of this study which is lacking in the manuscript.
Tables and Figures:
1. The legends of the figures are not crisp and not completely bringing out the sense of the figures. Rewrite it accordingly.
2. The placement of tables and figures in the manuscript should be done appropriately, which is missing in this manuscript.
Author Response
Reviewer’s comments to Authors-
- Grammatical errors are present, please revise the whole manuscript to remove any possible grammatical errors, redundancy and typos.
Author response: We tried to correct all grammatical and typing errors.
- The title of present study is fine however, the paper somewhat failed to answer some objectives of the study. This needs to be clearly addressed in the manuscript. Please revise it.
Author response: (1) This study showed that salt-tolerant rice such as Pokkali had better photosynthesis performance than the less tolerant genotypes and resulted in higher biomass under high salinity. (2) This study showed that the improved line CSSL8-94 which resulted from the introgression of drought-tolerant genes (located on DT-QTL8) into KDML105 genome, the line RD73 (introgressed with SKC1 salt-tolerant gene) had better growth in the heavy-saline field than KDML105. Moreover, the improved line TSKC1-144 (which carried both drought- and salt-tolerant genes) showed even higher growth than CSSL8-94 and RD73. This data on the benefit of pyramiding the two genes is first reported in this study. These improved lines proved to be good potential sources of breeding materials for rainfed lowland rice areas which are affected by salinity.
- Error in sentence formation, please revise the whole manuscript to avoid the use of long sentences and some paragraphs are very short. Paraphrasing is also required in the manuscript.
Author response: We have revised and modified the sentence formation, and tried not to use long sentences, the short paragraph (the first paragraph of discussion) was corrected.
- Please maintain the uniformity while in-text citation and referencing.
Author response: We have carefully revised all in-text citation and referencing
- In the keywords, it is strongly advisable use suitable words that can aid in finding out the manuscript in current registers or indexes. Strictly avoid the use of title words in the keywords.
Author response: The revised ‘Keywords’ are diurnal chlorophyll fluorescence, drought-tolerance QTL, Oryza sativa L., SKC1 gene, soil salinity
- The beginning of a new paragraph should be after some space and avoid beginning of new sentences with abbreviation, check in complete manuscript.
Author response: Revised and corrected.
- Uniformity in referencing is missing from the manuscript. Revise it.
Author response: We have carefully revised all references.
- It is recommended that authors should remove the below listed irrelevant self-cited papers from the manuscript-
- Santanoo, S.; Vongcharoen, K.; Banterng, P.; Vorasoot, N.; Joyloy, S.; Roytrakul, S.; Theerakulpisult, P. Seasonal variation in diurnal photosynthesis and chlorophyll fluorescence of four genotypes of cassava (Manihot esculenta Crantz) under irrigation conditions in a tropical savanna climate. Agronomy 2019, 9, 206.
- Pamuta, D.; Siangliw, M.; Sanitchon, J.; Pengrat, J.; Siangliw, J.L.; Toojinda, T.; Theerakulpisut, P. Photosynthetic performance in improved ‘KDML105’ rice (Oryza sativa L.) lines containing drought and salt tolerance genes under drought and salt stress. Pertanika J. Trop. Agric. Sci. 2020, 43, 653–675.
- Kanawapee, N.; Sanitchon, J.; Srihaban, P.; Theerakulpisut, P. Genetic diversity analysis of rice cultivars (Oryza sativaL.) differing in salinity tolerance based on RAPD and SSR markers. Electron. J. Biotechnol. 2011, 14, 1–17.
- Nounjan, N.; Wuttipong Mahakham, W.; Siangliw, J.L.; Toojinda, T.; Theerakulpisut, P. Chlorophyll retention and high photosynthetic performance contribute to salinity tolerance in rice carrying drought tolerance quantitative trait Loci (QTLs). Agriculture 2020, 10, 620.
Author response: We have removed Santanoo et al. (2019), Kanawapee et al. (2011) and Nounjan et al. (2020) as suggested. But we need to keep Pamuta et al. (2020) because this work reported photosynthesis performance (in pot experiment) of two genotypes (KDML105 and CSSL8-94) that were also investigated in this manuscript (field experiment).
Highlights, Abstract and Introduction:
- Highlights section should be added; significant results must be included in the highlight section of this manuscript, adding a graphical abstract would also be helpful for summarizing the importance of this study.
Author response: This journal does not require the ‘Highlight’ section and graphical abstract.
- Novelty statement is also omitted from the manuscript which is important for emphasizing the exclusivity of your study.
Author response: We showed the beneficial effects of gene pyramiding that the improved line TSKC1-144 which carries both drought tolerance QTL and SKC1 salt-tolerance gene in the genetic background of KDML105 had better growth under heavy-saline field than the lines which carries only drought tolerance QTL (i.e., CSSL8-94) or only SKC1 gene (RD73). The genotype with the highest salt tolerance level i.e. Pokkali had the best photosynthetic performance under salinity resulting in enhanced growth.
- Add significant results from all the sections precisely in the abstract.
Author response: Due to limited number of words, we tried to include as much significant results as possible. Now the revised abstract contained significant results including photosynthesis, chlorophyll fluorescence, biomass, comparison among genotypes, and comparison among non-saline VS saline fields.
- ‘‘Leaf greenness (SPAD values), on the other hand, remained high from the early vegetative to milky stage then reduced at maturity for all three conditions.” What does ‘for all three conditions’ signifies here, what conditions are being discussed is not clear in this line, please clarify.
Author response: This was corrected to ‘….all three field conditions.”
- “Rice response to salt stress is complex and depends on the type of salt stress, duration of salt exposue, and growth stage”, rephrase the sentence as it should be ‘’response of rice under salt stress….’, and correct the spelling of ‘exposure’.
Author response: This sentence has been corrected as suggested.
- “Accumulation of toxic Na+in plant tissues disturbs metabolic processes, particularly photosynthesis, and all major morpho-physiological and yield-related traits including tiller number, panicle length, spikelet number per panicle”, toxic Na+ isn’t the correct term, either mention the toxic level for Na+ or rephrase the sentence.
Author response: “Accumulation of toxic Na+ …” has been changed to “Accumulation of Na+ to toxic levels in plant tissues……”
- “Immediate effects of salinity on photosynthesis of rice involved an induction of stomatal closure, due to salt-induced osmotic stress”, use the term “photosynthetic ability” instead of “photosynthesis”.
Author response: Already changed as suggested.
- ‘‘This information will help determine genotypic potential and/or environmental factors limiting photosynthetic performance, and growth of rice. This information is expected to be useful for providing a set of guidelines for improving management of cultural practice for rice growing in saline fields.” This paragraph will be more suitable in the Conclusion section.
Author response: We did not remove this part from the ‘Introduction’ because we want the readers to perceive the usefulness of this work. And then we added the future perspectives obtained from our results in the ‘conclusion’.
Materials and Methods:
- ‘‘Seed germination was started on 18 June 2021, and seedlings were cultured at Khon Kaen Rice Research Center. When the plants were 40 days old, they were up--rooted and transplanted in the ploughed fields.’’ It is not clear whether the seed germination was done in pots or not, and how many replicates were planted, please specify.
Author response: Seed germination and seedling growth were done in a large cement block (1.5 m x 2.0 m) filled with paddy soils. For each genotype, 50 g (approximately 2,000 seeds) were germinated.
- ‘‘In early rainy season (May, 2021), soil EC of all plots were decreased from the hot season approximately 2.53, 2.87 and 0.12 for heavy-saline, semi-saline and non-saline plot, respectively’’. Does this constantly changing salinity of the soil due to varying weather conditions had any impact on the experiment, please explain.
Author response: The weather conditions especially the rainfall and temperature impose some effects on soil salinity. Rice production in slightly saline areas is higher in the years with higher rainfalls than dryer years due to dilution effects. The hotter year will result in lower rice production due to more evaporation and hence higher salinity.
Results and Discussion:
- During the hot season (March to April), prior to the rice growing season, the mean total rainfall (114.1 mm). What does ‘mean total rainfall’ means? either it is total rainfall or it could be the mean of the total rainfall, please specify.
Author response: ‘mean total rainfall’ has been changed to ‘’the mean of total rainfall”. Total rainfall in March was 92.3 mm, and in April was 135.8 mm, therefore ‘the mean of total rainfall’ was 114.1 mm.
- “Total rainfall, number of rainy days, monthly mean relative humidity (RH) and air temperature (T) of each month in 2021 were presented in Figure 1.” Why the complete ‘Result’ section is written in past tense when it should be in present tense, for example this sentence must be written as Total rainfall, number of rainy days, monthly mean relative humidity (RH) and air temperature (T) of each month in 2021 arepresented in Figure 1. The use of ‘were’ should be replaced with ‘are’.
Author response: Corrected as suggested.
- “The measurements at the flowering stage were conducted on flag leaves at the end of rainy season 21 days after water drainage from the rice fields and soil EC were highest.” How come the EC was highest after rainfall and water drainage as the water drainage after rainfall would result in the declining of salts present in the soil due to leaching, please explain.
Author response: The soil EC was high after water drainage (Figure 1D) because the majority of salts are present in the soil rather than in the flooded water. Moreover, as soil water evaporated more salts from deeper soil layers will move up to the soil surface (See Figure S2 in the Supplementary for salt crust in the dry season)
- “At maturity, the means plant height across genotypes in the non-, semi-, and heavy-saline fields were 166.2, 146.95, and 136.15 cm,”, this should be meanplant height, nor means.
Author response: Corrected as suggested.
- The section of the results in which photosynthetic parameters are discussed is extended unnecessarily, this section should be rewritten.
Author response: We have revised the ‘Results’ section and modified some sentences which were difficult to understand.
- Discussion section must more studies from previous reports as evidence for your study, recent studies must be included.
Author response: We added 5 more references i.e. [35], [36], [45], [50], [51]
- The authors failed to explain all of their findings in the discussion section with specific mechanisms. It is advisable to maintain balance between the result and the discussion section.
Author response: We added more discussion regarding (1) the improvement of growth under salinity of the improved lines introgressed with DT-QTL8 and SKC1 over that of the recipient KDML105, (2) the relationship between electron transport rate and Pn (CO2 fixation) under stress conditions, (3) the mechanism of HKT1;5 transporter protein which was encoded by the SKC1 gene
References:
There are lots of statements/sentences throughout the manuscript that essentially require proper validation and citation with previous studies which are largely missing in the present manuscript. Introduction, Result, and Discussion sections poorly cited with the references and strongly recommended to update and validation with previous studies. Therefore, some of the papers listed below should be considered and cited appropriately in the Introduction, Result, and Discussion sections of this manuscript which will certainly upgrade and enhance the Ms. quality significantly.
- Gupta, S., et al. (2021). Salicylic acid alleviates chromium (VI) toxicity by restricting its uptake, improving photosynthesis and augmenting antioxidant defense in Solanum lycopersicumL. Physiology and Molecular Biology of Plants, 27(11), 2651-2664.
- Yadav, M., et al. (2022). Foliar application of α-lipoic acid attenuates cadmium toxicity on photosynthetic pigments and nitrogen metabolism in Solanum lycopersicumL. Acta Physiologiae Plantarum, 44(11), 1-10.
- Gupta, P., et al. (2022). 24-Epibrassinolide Regulates Functional Components of Nitric Oxide Signalling and Antioxidant Defense Pathways to Alleviate Salinity Stress in Brassica junceaL. cv. Varuna. Journal of Plant Growth Regulation, 1-16.
- Prajapati, P., et al. (2022). Nitric oxide mediated regulation of ascorbate-glutathione pathway alleviates mitotic aberrations and DNA damage in Allium cepaL. under salinity stress. International Journal of Phytoremediation, 1-12.
Author response: We have added one of the suggested references (Gupta et al. 2022) which was most relevant to our work.
Conclusion:
- The conclusion section should be in accordance to the result and discussion section, random addition of results should be avoided.
- Conclusion section failed to enlighten the spirit of the finding and is missing the results. Revise it precisely.
- Conclusion section must also include the future perspectives of this study which is lacking in the manuscript.
Author response: We have improved ‘Conclusion’ – we removed the results from previous study, and added future perspective in regard to the use of the investigated lines in breeding.
Tables and Figures:
- The legends of the figures are not crisp and not completely bringing out the sense of the figures. Rewrite it accordingly.
Author response: We have improved the figure legends.
- The placement of tables and figures in the manuscript should be done appropriately, which is missing in this manuscript.
Author response: The manuscripts contain 7 Figures, and each was placed immediately after the description in the ‘Results’. The supplements contained 4 tables (Table S1, S2, S3 and S4) plus 2 figures (Fig. S1 and Fig S2 – newly added)
Round 2
Reviewer 3 Report
Dear Author and Editor
I have suggested four most relevant papers to upgrade the Ms quality but author considered only one out of four. This will compromise the Ms quality. I suggest the authors consider all remaining three papers. I am not satisfied with this version of Ms and it requires minor revision.